# Uncovering Conceptual Blindspots in Generative Image Models Using Sparse Autoencoders

**Matyas Bohacek**[1*]    **Thomas Fel**[2*]    **Maneesh Agrawala**[1]    **Ekdeep Singh Lubana**[3]

[1]Department of Computer Science, Stanford University
[2]Kempner Institute, Harvard University
[3]CBS-NTT Program in Physics of Intelligence, Harvard University

## Abstract

Despite their impressive performance, generative image models trained on large-scale datasets frequently fail to produce images with seemingly simple concepts—e.g., `human hands` or `objects appearing in groups of four`—that are reasonably expected to appear in the training data. These failure modes have largely been documented anecdotally, leaving open the question of whether they reflect idiosyncratic anomalies or more structural limitations of these models. To address this, we introduce a systematic approach for identifying and characterizing "conceptual blindspots"—concepts present in the training data but absent or misrepresented in a model's generations. Our method leverages sparse autoencoders (SAEs) to extract interpretable concept embeddings, enabling a quantitative comparison of concept prevalence between real and generated images. We train an archetypal SAE (RA-SAE) on DINOv2 features with $32,000$ concepts—the largest such SAE to date—enabling fine-grained analysis of conceptual disparities. Applied to four popular generative models (Stable Diffusion 1.5/2.1, PixArt, and Kandinsky), our approach reveals specific suppressed blindspots (e.g., `bird feeders`, `DVD discs`, and `whitespaces on documents`) and exaggerated blindspots (e.g., `wood background texture` and `palm trees`). At the individual datapoint level, we further isolate memorization artifacts — instances where models reproduce highly specific visual templates seen during training. Overall, we propose a theoretically grounded framework for systematically identifying conceptual blindspots in generative models by assessing their conceptual fidelity with respect to the underlying data-generating process. Code: `https://conceptual-blindspots.github.io`.

## 1 Introduction

Generative image models trained on large scale datasets have achieved unprecedented capabilities, allowing their use in applications both within the vision domain OpenAI (2024); Peebles and Xie (2023); Ramesh et al. (2021); Saharia et al. (2022); Nichol et al. (2021); Wang et al. (2024); Poole et al. (2022); Richardson et al. (2023); Rombach et al. (2022) and well beyond that Ahn et al. (2022); Huang et al. (2022a;b); Rombach et al. (2022); Chen et al. (2024); Zhong et al. (2024); Siddiqui et al. (2024). Despite this success, several qualitative (Marcus et al., 2022; Cabrera et al., 2021; Heigl, 2025) and quantitative studies (Liu et al., 2023a; Conwell et al., 2024) have shown that, at times, models can struggle to generate images with relatively simple concepts, e.g., `human hands` (Lu et al., 2024; Narasimhaswamy et al., 2024; Zhangli et al., 2024; Fallah et al., 2025), `objects appearing in groups of four` (Cao et al., 2025), and `negations or object relations` (Conwell and Ullman, 2022; Conwell et al., 2024). In fact, when prompted to generate images containing such concepts, models tend to produce outputs with related structures, but not precisely the ground truth concept—e.g., producing hands with six fingers. These failure modes, which we call "conceptual blindspots"[1], can be unintuitive, since one may reasonably expect models have had enough exposure

---

*Equal contribution. Email: `maty@stanford.edu`, `tfel@g.harvard.edu`, `maneesh@cs.stanford.edu`, `ekdeeplubana@fas.harvard.edu`.

[1]We borrow the term "blindspots" from psychology literature (Banaji and Greenwald, 2016), wherein it is used to refer to scenarios where an agent makes biased decisions despite exposure to observations that contradicts the rationale behind those decisions.

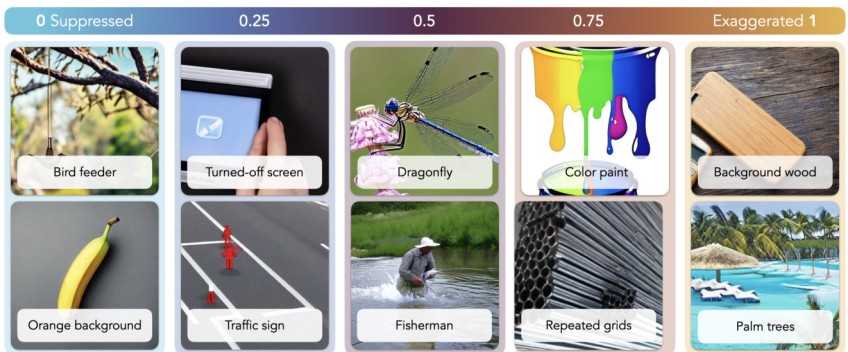

Figure 1: $\delta(k)$ quantifies a model's tendency to over- or under-generate a concept $c_k$ compared to its natural-data frequency. We deem concepts with $\delta(k) < 0.1$ as **suppressed conceptual blindspots** and concepts with $\delta(k) > 0.9$ as **exaggerated conceptual blindspots**. The depicted images, generated by four popular generative image models, show examples of conceptual blindspots as well as aligned concepts. The models are completely unable to generate suppressed blindspots (e.g., `bird feeder`), despite diverse prompting and steering strategies. For concepts with $\delta(k) \approx 0.25$ (e.g., `traffic sign`), the models exhibit substantial deficiencies. In contrast, exaggerated blindspots emerge unprompted, at rates far exceeding their distribution in natural images.

to demonstrations accurately detailing such concepts. This raises the question whether such failures reflect intriguing quirks of certain specific concepts, or whether they are demonstration of a more systematic phenomenon under which, for a broad spectrum of concepts, models fail to or are overly likely to produce images containing them.

Answering this question remains infeasible with existing approaches for evaluating generative image models Stein et al. (2023); Wang et al. (2023a). Specifically, existing approaches generally rely on coarse-grained measures that are meant to assess image realism, e.g., FID (Heusel et al., 2017), and hence do not capture distributional failures. Methods like CLIPScore evaluate generation diversity or distribution-coverage statistics (Hessel et al., 2021; Dombrowski et al., 2024; Hwang et al., 2024), hence offering partial insights to our question, but not at the granularity of fine-grained features or concepts Theis et al. (2015); Naeem et al. (2020), making it difficult to identify conceptual blindspots. Finally, qualitative analyses for evaluating generative models, such as participant surveys (Cabrera et al., 2021; Nichol et al., 2021; Petsiuk et al., 2022; Xu et al., 2023; Wu et al., 2023a) or open-ended exploration (Bau et al., 2019), can identify failures in models' ability to capture certain concepts, but do not offer a scalable methodology that would allow for comparability.

**This work.** Motivated by the above, we argue identifying and analyzing conceptual blindspots in a generative image model requires designing a methodology that, in an automated and unsupervised manner, can elicit concepts in the data distribution that have a mismatch between their odds of generation by the true data-generating process versus the trained model. Our contributions include:

- **Formalizing Conceptual Blindspots in Generative Image Models.** We introduce a systematic framework for identifying and quantifying conceptual blindspots in generative image models compared to natural images (Sec. 2). This formalization moves beyond anecdotal or human-defined concept evaluations, offering a principled approach to understand the models' limitations.

- **A Scalable, Unsupervised Approach for Identifying Conceptual Blindspots using Sparse Autoencoders.** We develop a structured methodology using sparse autoencoders (SAEs) to extract and compare concept distributions between natural and synthesized images (Sec. 3). To do so, we employ SAEs, which decompose the high-dimensional activation space of models into sparse, human-interpretable concepts; namely, they are trained to reconstruct model activations using a sparse combination of learned feature directions (concepts). Each concept can then be assigned a human-interpretable label through *autointerpretability*: examining high-activating exemplars and prompting an LLM to describe them. To this end, we train and open-source an archetypal SAE (RA-SAE) on DINOv2 features with $32,000$ concepts, the largest such RA-SAE to date.

- **Exploratory Tool and Analysis.** Our exploratory web tool enables both distribution- and datapoint-level analysis of blindspots across models (Sections 4.1-4.4). We apply our method to four popular models (Section 4) and identify specific instances of both suppressed conceptual blindspots (e.g., `bird feeders`, `DVD discs`, and `whitespaces on documents`) and exaggerated conceptual blindspots (e.g., `wood background texture` and `palm trees`), shown in Fig. 1.

## 2 FORMALIZING CONCEPTUAL BLINDSPOTS IN GENERATIVE MODELS

We begin by formalizing the notion of *conceptual blindspots*: systematic discrepancies between the conceptual content of natural images and that of model-generated outputs. This formulation enables us to derive principled, quantitative measures that characterize which concepts are under or over represented by a generative model relative to its data distribution. The process is illustrated in Fig. 2. While we rely on standard assumptions in this pursuit (Von Kugelgen et al., 2021; Locatello et al., 2019; Zimmermann et al., 2021; Gresele et al., 2020; 2021), empirically we find meaningful phenomenology is elicited even when these assumptions are violated.

**Definition 1** (**Data-Generating Process**). *Let $\mathcal{C} \subset \mathbb{R}^K$ denote a latent space with a Boltzmann prior $p(\boldsymbol{c}) = \exp(-E(\boldsymbol{c}))Z^{-1}$, where $E(\cdot)$ denotes an energy function that linearly decomposes over individual latents and $Z$ is the corresponding partition function, i.e., $E(\boldsymbol{c}) = \sum_k E(c_k)$ and hence $p(\boldsymbol{c}) = \prod_k p_k(c_k)$. A* data-generating process (DGP) *is an invertible function $\boldsymbol{G} : \mathcal{C} \to \mathcal{X}$ that maps the latents $\boldsymbol{c} \in \mathcal{C}$ to observations $\boldsymbol{x} \in \mathcal{X}$, i.e., $\boldsymbol{x} = \boldsymbol{G}(\boldsymbol{c})$.*

For notational simplicity, we use $p(\cdot)$ to denote both the latent density $p(\boldsymbol{c})$ and its push-forward to image space $p(\boldsymbol{x})$, where $\boldsymbol{x} = \boldsymbol{G}(\boldsymbol{c})$. This is justified by the invertibility of $\boldsymbol{G}$, which induces a valid distribution over $\mathcal{X}$ via the change of variables formula.

In essence, the individual dimensions of the latent space reflect the **Concepts** underlying the data-distribution $P_{\mathcal{X}}$, defined over some observation space of images $\mathcal{X}$. For example, different latents may correspond to concepts like color, shape, size, location, and so on (Okawa et al., 2023; Park et al., 2024). We also let the data-generating process associate a text-description $\boldsymbol{t} \in \mathcal{T}$ with any image sampled from the data distribution, but do not explicitly model it. These text descriptions can then be used to train a text-conditioned **Generative image model** $g_{\boldsymbol{\theta}}$, with parameters $\boldsymbol{\theta}$, on a set of image-text pairs to map a noise signal $\eta \sim \mathcal{N}(\boldsymbol{0}, \boldsymbol{I})$ and a text-description of the scene $\boldsymbol{t}$ to produce images $\boldsymbol{x}$ illustrating the latter.

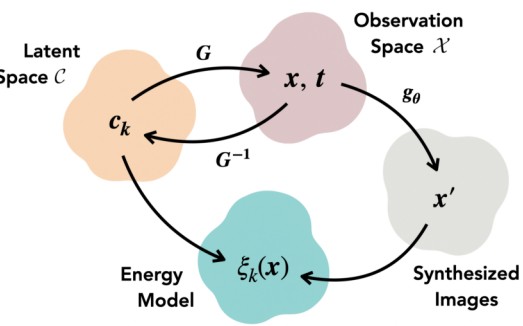

Figure 2: **Intuitive mapping of our framework.** Latent concepts $c_k \in \mathcal{C}$ are mapped to observations $(\boldsymbol{x}, \boldsymbol{t})$ through the (invertible) DGP. $g_{\boldsymbol{\theta}}$ generates images $\boldsymbol{x}'$ based on $\boldsymbol{t}$. The energy model $\xi_k(\boldsymbol{x})$ extracts concept representations from both $\boldsymbol{x}$ and $\boldsymbol{x}'$, enabling comparison of concept distributions to identify blindspots.

To define conceptual blindspots in the model $g_{\boldsymbol{\theta}}$, we must assess the probability mass assigned to a concept by the data-generating process, comparing it to the mass assigned by the model. To this end, we take an evaluation set of natural image-text pairs $(\mathcal{D}_{\mathcal{X}}, \mathcal{D}_{\mathcal{T}})$ and define a set of synthetically generated images $\mathcal{D}'_{\mathcal{X}}$ using the text descriptions. To estimate the probability of occurrence of a concept, we consider an **Energy model** $\xi : \mathcal{X} \to \mathbb{R}^d$ that maps images to a $d$-dimensional vector, where each dimension associates a scalar representing the energy in the $k^{\text{th}}$ concept, i.e., $\xi_k(\boldsymbol{x}) = E(c_k)$. These estimates are feasible because we assume the data-generating process is invertible. Correspondingly, the probability the data-generating process associates with the image $\boldsymbol{x}$ can then be defined as $p_k(\boldsymbol{x}) \propto \exp(-\sum_k \xi_k(\boldsymbol{x}))$ (where $Z_{\xi}$ is the partition function), hence yielding a population-level estimate $p_k(\mathcal{D}_{\mathcal{X}}) = \prod_{\boldsymbol{x} \in \mathcal{D}_{\mathcal{X}}} p_k(\boldsymbol{x})$. Using this and the sigmoid map $\sigma(\cdot)$, we define below the average energy difference in the $k^{\text{th}}$ concept between the datasets $\mathcal{D}_{\mathcal{X}}, \mathcal{D}'_{\mathcal{X}}$.

**Definition 2** (**Energy Difference**). *Let $\boldsymbol{x} \in \mathcal{D}_{\mathcal{X}}$ denote a real image sampled from the data-generating process $\boldsymbol{G}$, and let $\boldsymbol{x}' \in \mathcal{D}'_{\mathcal{X}}$ be a synthetic image generated by the model $g_{\boldsymbol{\theta}}$. Let $\xi_k : \mathcal{X} \to \mathbb{R}$ denote the energy assigned to the $k^{th}$ concept by the energy model $\xi$. We define the energy difference for concept $k$ as:*

$$\delta_{g_{\boldsymbol{\theta}} \leftrightarrow \boldsymbol{G}}(k) = \sigma\left(\mathbb{E}_{\boldsymbol{x}'}(\xi_k(\boldsymbol{x}')) - \mathbb{E}_{\boldsymbol{x}}(\xi_k(\boldsymbol{x}))\right)$$
$$= \frac{p_k(\mathcal{D}'_{\mathcal{X}})}{p_k(\mathcal{D}_{\mathcal{X}}) + p_k(\mathcal{D}'_{\mathcal{X}})}, \qquad (1)$$

*where the expectations are taken over $\mathcal{D}'_{\mathcal{X}}$ and $\mathcal{D}_{\mathcal{X}}$, respectively, and $p_k(\mathcal{D}) \propto \exp\left(-\sum_{\boldsymbol{x} \in \mathcal{D}} \xi_k(\boldsymbol{x})\right)$ denotes the unnormalized conceptual probability mass of dataset $\mathcal{D}$ under concept $k$.*

Thus, the energy difference in the $k^{\text{th}}$ concept describes the ratio of the probability a concept occurs in a set of observations (here, $\mathcal{D}'_{\mathcal{X}}$) compared to a baseline dataset (here, $\mathcal{D}_{\mathcal{X}}$). Based on this measure, we can now define conceptual blindspots as follows.

**Definition 3** (**Suppressed / Exaggerated Conceptual Blindspots**). *Given a generative image model $g_{\boldsymbol{\theta}}$, we say, compared to the data-generating process $\boldsymbol{G}$, $c_k$ is a* suppressed *conceptual blindspot in the model if $\delta_{g_{\boldsymbol{\theta}} \leftrightarrow \boldsymbol{G}}(k) < \lambda_{\min}$ and* exaggerated *if $\delta_{g_{\boldsymbol{\theta}} \leftrightarrow \boldsymbol{G}}(k) > \lambda_{\max}$.*

Overall, we define a conceptual blindspot as a concept whose likelihood of occurrence in generated images deviates markedly, either through suppression or exaggeration, from its prevalence under the data-generating process. Suppressed concepts exhibit disproportionately low activation (e.g., $\delta(k) < \lambda_{\min}$), whereas exaggerated concepts are overrepresented (e.g., $\delta(k) > \lambda_{\max}$). Throughout our analysis, we adopt threshold values of $\lambda_{\min} = 0.1$ and $\lambda_{\max} = 0.9$ to isolate these regimes.

We also note this definition is related to the idea of "mode collapse" studied in past work (e.g., see Bau et al. (2019)): the difference is in the granularity at which the analysis is performed. Specifically, mode collapse focuses on exaggerated / suppressed odds of generating *entire images*, while we focus on changed odds of specific concepts. For example, if a model fails to produce images of an object with a `white background`, we say this concept is a suppressed conceptual blindspot.

## 3 METHOD: OPERATIONALIZING THE DEFINITION OF CONCEPTUAL BLINDSPOTS

We next discuss our pipeline for identifying conceptual blindspots in a generative model $g_{\boldsymbol{\theta}}$. As per Sec. 2, the salient objects we need for this are (i) a set of images sampled from $g_{\boldsymbol{\theta}}$ that allow comparison with the ground-truth generative process, and (ii) an energy model which enables said comparison. Below, we use $\| \cdot \|_F$ to denote the Frobenius norm and $\| \cdot \|_0$ to denote the number of non-zero entries (the $\ell_0$ pseudo-norm). For a vector or matrix $\boldsymbol{X}$, $\boldsymbol{X} \geq 0$ implies element-wise non-negativity. For $n > 0$, we let $[n] := \{1, \ldots, n\}$, and denote the $i$-th row of a matrix $\boldsymbol{A}$ by $\boldsymbol{A}_i$.

**From Prompts to Latent Representations.** To identify conceptual blindspots in a model $g_{\boldsymbol{\theta}}$, we compare a dataset $\mathcal{D}_{\mathcal{X}}$ of image-caption pairs $(\boldsymbol{x}, \boldsymbol{t})$ sampled from the data-generating process $\boldsymbol{G}$ and their synthetic counterparts sampled from the generative model $g_{\boldsymbol{\theta}}$ using the text descriptions.

Specifically, given $\boldsymbol{t}$, we synthesize a counterpart image $\boldsymbol{x}'$ using a pretrained text-to-image generator $g_{\boldsymbol{\theta}} : \mathcal{T} \to \mathcal{X}$, implemented as a denoising diffusion probabilistic model (DDPM) (Razzhigaev et al., 2023; Stability AI, 2022; Chen et al., 2023a). Sampling occurs in latent space via a reverse trajectory $(\boldsymbol{\gamma}_t)_{t=0}^{T}$:

$$\boldsymbol{\gamma}_T \sim \mathcal{N}(\boldsymbol{0}, \boldsymbol{I}), \qquad \boldsymbol{\gamma}_{t-1} = \frac{1}{\sqrt{\alpha_t}} \big( \boldsymbol{\gamma}_t - \frac{1-\alpha_t}{\sqrt{1-\bar{\alpha}_t}} \, \boldsymbol{\varepsilon}_{\boldsymbol{\theta}}(\boldsymbol{\gamma}_t, t, \boldsymbol{c}) \big) + \sigma_t \boldsymbol{\eta}_t, \quad \text{and} \quad \boldsymbol{\eta}_t \sim \mathcal{N}(\boldsymbol{0}, \boldsymbol{I}),$$

where $\alpha_t \in (0, 1)$ and $\bar{\alpha}_t = \prod_{s=1}^{t} \alpha_s$ follow the standard cosine noise schedule. The final latent $\boldsymbol{\gamma}_0$ is decoded via a pretrained VAE to yield the synthetic image $\boldsymbol{x}' = \text{VAE}(\boldsymbol{\gamma}_0)$. For the remainder of the paper, we treat $g_{\boldsymbol{\theta}}$ as a black box that maps prompts to images: $\boldsymbol{t} \mapsto \boldsymbol{x}'$.

**Defining the Energy Model.** Building on prior work that shows the ability of self-supervised learning methods to invert the data-generating process and identify the energy function underlying it up to linear transformations (Zimmermann et al., 2021; Von Kugelgen et al., 2021; Khemakhem et al., 2020; Hyvarinen et al., 2019), we use DINOv2 (Oquab et al., 2023) for our analysis[2]. Under the expectation that the number of concepts underlying the DGP is larger than the dimensionality of the model's feature space (Elhage et al., 2022; Bricken et al., 2023), we train sparse autoencoders (SAEs) on its features to identify subspaces corresponding to these concepts (Fel et al., 2025; Cunningham et al., 2023; Gao et al., 2025; Templeton et al., 2024; Rajamanoharan et al., 2024). The intuition here is that if the concepts underlying the generative process are modeled via approximately orthogonal

---

[2]We use DINOv2 in our energy model because its self-supervised training on large-scale unlabeled data yields emergent, highly structured visual representations that capture broad semantic and geometric regularities without text supervision. These embeddings have proven robust across tasks (classification, segmentation, depth estimation, tracking) and domains (natural, medical, satellite imagery) Oquab et al. (2023); Liu et al. (2023b); Ayzenberg et al. (2024).

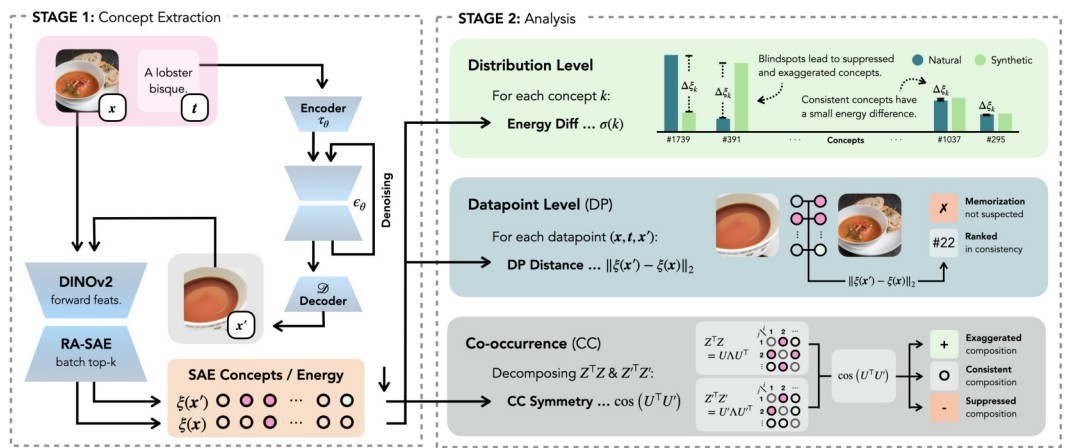

Figure 3: **Concept Extraction Pipeline.** For a triplet $(\boldsymbol{x}, \boldsymbol{t}, \boldsymbol{x}')$, the concepts in $\boldsymbol{x}$ and $\boldsymbol{t}$ are extracted by obtaining each image's DINOv2 features, which are further processed by a RA-SAE into sparse concept embeddings, yielding energy models $\xi(\boldsymbol{x})$ and $\xi(\boldsymbol{x}')$, respectively. In particular, $\xi_k(\boldsymbol{x}) = E(c_k)$ holds the energy in the $k^{\text{th}}$ concept.

directions by DINOv2 (as assumed in our independence constraint in Def. 1), then an SAE should be able to isolate these concepts along individual dimensions in its latent space (Elhage et al., 2022). The activation associated by the SAE to a dimension will serve as our approximation of the ground-truth energy function assigned to the concept modeled by that dimension.

Formally, using $\boldsymbol{f} : \mathcal{X} \to \mathbb{R}^d$ to denote our feature extraction module (i.e., the DINOv2 model), we extract features $\boldsymbol{a} = \boldsymbol{f}(\boldsymbol{x}) \in \mathbb{R}^d$ from both natural and synthetic images from datasets $\mathcal{D}_\mathcal{X}, \mathcal{D}'_\mathcal{X}$. Assuming the count of image-text pairs is $n$, we stack the real and generated features into matrices $\boldsymbol{A}, \boldsymbol{A}' \in \mathbb{R}^{n \times d}$. We then decompose each feature vector into a sparse combination of learned concept atoms using an SAE. Specifically, let $\boldsymbol{D} \in \mathbb{R}^{d \times K'}$ denote a dictionary of $K'$ concept vectors, and let $\boldsymbol{\Psi} : \mathbb{R}^d \to \mathbb{R}^{K'}$ be the SAE encoder that maps input features to sparse codes. Applying $\boldsymbol{\Psi}$ row-wise yields the matrix of activations $\boldsymbol{Z} = \boldsymbol{\Psi}(\boldsymbol{A}) \in \mathbb{R}^{n \times K'}$, where each row $\boldsymbol{z}_i = \boldsymbol{\Psi}(\boldsymbol{a}_i)$ represents the concept decomposition of an image. The decoder reconstructs features via $\boldsymbol{Z}\boldsymbol{D}^\top$, and the SAE is trained to minimize the reconstruction error subject to sparsity and non-negativity:

$$\min_{\boldsymbol{\Psi}, \boldsymbol{D}} \left\| \boldsymbol{A} - \boldsymbol{\Psi}(\boldsymbol{A})\boldsymbol{D}^\top \right\|_F^2 \quad \text{s.t.} \quad \boldsymbol{\Psi}(\boldsymbol{A}) \geq 0, \ \|\boldsymbol{\Psi}(\boldsymbol{A})_i\|_0 \ll K' \quad \forall i \in [n]. \tag{2}$$

Vanilla SAEs often drift toward arbitrarily oriented dictionaries, making downstream analyses highly sensitive to the random seed. To mitigate this instability and make our study reproducible and independent of the seed, we employ the *Archetypal* SAE (RA-SAE) Fel et al. (2025) on a TOP-K sparsity constraint Gao et al. (2025). RA-SAE constrains the dictionary $\boldsymbol{D}$ to be a convex combination of training data. Specifically, we write $\boldsymbol{D} = \boldsymbol{W}\boldsymbol{A}$ with $\boldsymbol{W} \in \Omega_{K',n}$, the set of row-stochastic matrices in $\mathbb{R}^{K' \times n}$:

$$\Omega_{K',n} := \left\{ \boldsymbol{W} \in \mathbb{R}^{K' \times n} \mid \boldsymbol{W} \geq 0, \ \boldsymbol{W}\mathbf{1} = \mathbf{1} \right\}. \tag{3}$$

Thus every atom $\mathbf{D}_i$ lies in the convex hull of the data $\text{conv}(\mathbf{A})$, and any reconstruction $\boldsymbol{Z}\boldsymbol{D}^\top$ resides inside the conic hull of the data $\text{cone}(\mathbf{A})$. This ensures learned concepts remain faithful to the support of the data distribution (Fel et al., 2025). Once trained, the SAE provides a consistent set of sparse codes: $\boldsymbol{Z}$ for real images and $\boldsymbol{Z}'$ for their generated counterparts. *These codes capture the same prompt-conditioned visual semantics in terms of shared, interpretable concepts, with the activation value of the concept serving as energy values for our analysis of conceptual blindspots.* In summary then, our method defines a structured pipeline that, given a prompt and its associated real image $(\boldsymbol{t}, \boldsymbol{x})$, produces two sparse concept vectors $(\boldsymbol{z}, \boldsymbol{z}')$, enabling direct comparison of the real and generated visual content in a common conceptual basis.

This summarizes our full pipeline: starting from a captioned image $(\boldsymbol{t}, \boldsymbol{x})$, we synthesize a counterpart $\boldsymbol{x}'$ and map both images into a shared, sparse concept space via a vision encoder and a trained SAE, yielding interpretable representations $(\boldsymbol{z}, \boldsymbol{z}')$ that will serve as the foundation for evaluating conceptual shifts induced by the generative process.

# 4 RESULTS

We analyze four generative image models trained on LAION-5B—SD 1.5, SD 2.1, PixArt, and Kandinsky—using $|\mathcal{X}| = 10{,}000$ image-text pairs and their corresponding generations (Appendix N). Our analysis spans three levels (Fig. 3): a ● **distribution-level** evaluation reveals suppressed and exaggerated concepts; a ● **datapoint-level** analysis surfaces failures tied to ambiguity, omission, and memorization; and a ● **compositional** analysis uncovers subtle distortions in concept co-occurrence geometry. Our core contribution is an interactive exploratory tool, shown in Appendix C, from which all subsequent analyses emerge. Rather than exhaustively studying one phenomenon, we present high-level findings that highlight the tool's versatility and enable broader, customizable exploration.

## 4.1 THE HEAVY TAIL OF SUPPRESSED CONCEPTS

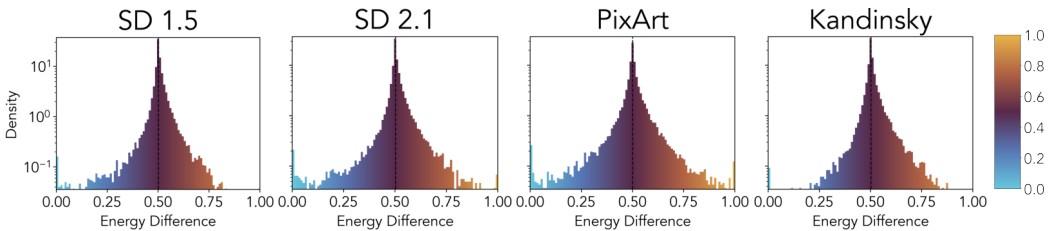

Figure 4: ● **Concept Energy Distribution.** Log-scale histograms of energy differences $\delta(k)$ across $32{,}000$ concepts, comparing the natural and synthesized distributions for each evaluated model. Values left of $0.5$ represent suppressed concepts (under-represented); values right of $0.5$ represent exaggerated concepts (over-represented).

To assess disparities between the generative models and the natural image distribution, we begin by analyzing the marginal energy difference $\delta(k)$ across 32,000 concepts learned using RA-SAE. As defined in Sec. 3, this quantity reflects the relative prevalence of each concept in the synthesized versus natural image sets. A value of $\delta(k) < 0.1$ indicates that concept $k$ is under-represented (suppressed) in the generated images, while $\delta(k) > 0.9$ indicates over-representation (exaggerated). Fig. 4 presents the distribution of $\delta(k)$ for each of the four evaluated models. Across all models, we observe heavy-tailed histograms with substantial mass on both extremes, suggesting systematic discrepancies in concept coverage. Notably, the left tail—corresponding to suppressed concepts—is denser and longer than the right, indicating a consistent tendency of concept suppression. This asymmetry is reflected in the negative skewness of the distributions: Skewness $= -0.54$ for SD 2.1, $-0.40$ for both SD 1.5 and PixArt, and $-0.23$ for Kandinsky.

We also note that while all models exhibit both suppressed and exaggerated concepts, their specific profiles differ. For instance, PixArt shows a wider spread, suggesting a more suppressed concept distribution. Nevertheless, the consistent left-skew in all distributions underscores a common tendency toward concept omission, though the specific characteristics of this behavior require further analysis, which we explore in the next Sections.

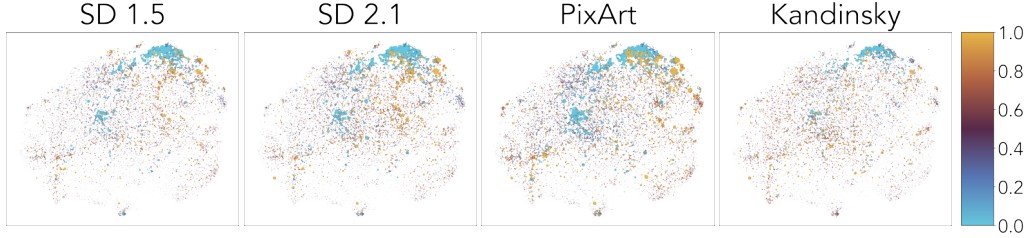

Figure 5: ● **Structure of Concept Energy Differences.** UMAP visualizations of $32{,}000$ concepts, colored according to their energy difference $\delta(k)$ between the natural and synthesized distributions. Clusters reveal patterns of conceptual blindspots, with suppressed concepts on the blue end and exaggerated ones on the yellow end.

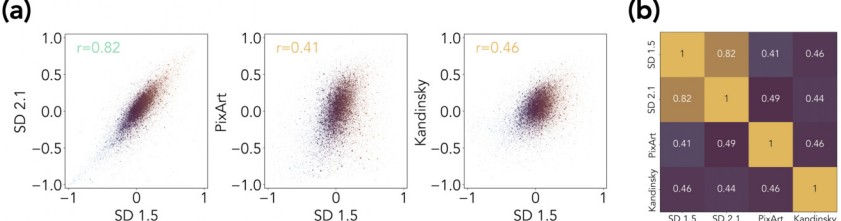

Figure 6: ● **Cross-Model Concept Energy Correlation.** Pairwise scatter plots of $\delta(k)$ across all four evaluated models, with Pearson correlation coefficients reported top left. Strong alignment between SD 1.5 and SD 2.1 contrasts with weaker correlations among other architectures, indicating model-specific blindspots. ● **Correlation Matrix of Conceptual Blindspots.** Heatmap of pairwise Pearson correlation coefficients for $\delta$ between all models, quantifying the degree of shared conceptual blindspots across these models.

## 4.2 STRUCTURE AND SPECIFICITY OF CONCEPTUAL BLINDSPOTS

While the previous section quantified marginal discrepancies in concept frequency, here we investigate their global structure by embedding the full set of 32,000 concepts into two dimensions using UMAP on the sparse codes, coloring the concepts by their $\delta(\cdot)$ values. As shown in Fig. 5, distinct clusters of concepts emerge across all models. These clusters often correspond to contiguous regions of conceptual blindspots, especially for suppressed (blue) concepts, suggesting that blindspots are quite structured—reflecting shared biases in either training distributions or architectural priors. To assess the consistency of these blindspot patterns across models, we further analyze both the magnitude and structure of concept-level $\delta(\cdot)$ values. Fig. 6 presents scatter plots and pairwise Pearson correlation coefficients between the $\delta(k)$ vectors of SD 1.5 and all other models. As expected, SD 1.5 and 2.1 exhibit strong correlation ($r = 0.82$), reflecting their shared architectural and training pipelines. In contrast, their correlations with PixArt and Kandinsky are substantially lower—$r = 0.41$ and $r = 0.46$, respectively—indicating that these models emphasize different regions of the conceptual space.

Overall, the analysis above reveals that while some blindspots are universally shared—likely due to properties of the dataset—others are highly model-specific, emerging from idiosyncrasies in training dynamics or model capacity. This motivates the need to identify and study both blindspots that are shared across models and ones that are unique to specific models in subsequent sections.

## 4.3 QUALITATIVE BLINDSPOT EXAMPLES

We next visualize specific examples of both suppressed and exaggerated blindspots to gauge what concepts fall under these regimes. Specifically, in Fig. 8a we show an example of a conceptual blindspot suppressed by all models—we find all evaluated models fail to reproduce the concept `solid white on documents`. As can be seen in the figures, despite the caption explicitly referencing this concept, none of the generated images reflect the intended visual semantics, suggesting that this region of the concept space is systematically under-sampled across models. Meanwhile, Fig. 8b highlights a model-specific blindspot: the concept `pan` is accurately captured by three models, yet conspicuously missing from generations produced by Kandinsky. This reinforces the findings from Sec. 4.1, where cross-model agreement was found to be high in some cases but limited in others.

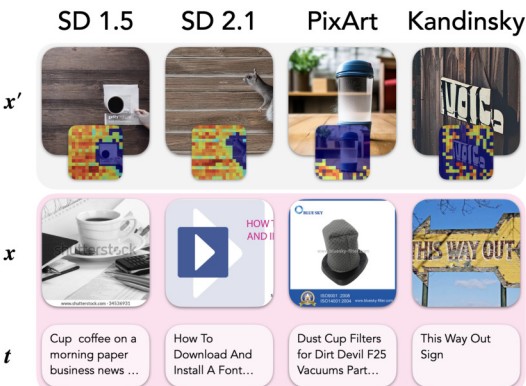

Figure 7: ● **Example of an Exaggerated Conceptual Blindspot.** Four synthesized images $x'$ with the `wooden surfaces and everyday objects` concept are shown alongside the corresponding natural image $x$ and text prompt $t$. As expected with exaggerated blindspots, the concept is prominent in $x'$ but absent in $x$ and $t$.

Conversely, in Fig. 7 we present a case of exaggeration, where the concept `shadow under animal` is overly emphasized in generated images. While shadows are mildly plausible, their consistent and pronounced rendering across models, relative to the more nuanced and variable occurrences in

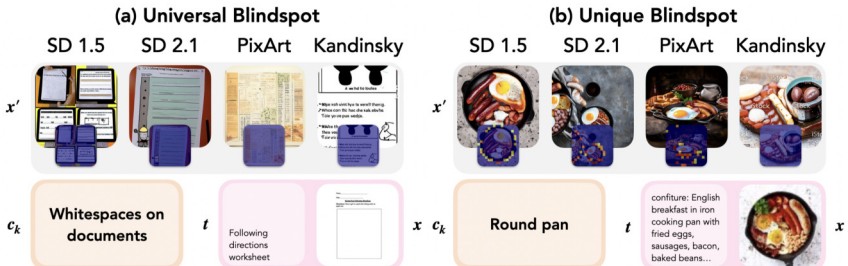

Figure 8: ● **Examples of Suppressed Conceptual Blindspots.** The natural images $x$, representative $c_k$ and $t$, shown alongside four synthesized images $x'$, generated using $S_\theta$. The universal blindspot is present in all evaluated models; the unique blindspot is only present in Kandinsky.

natural images, suggests an overactive prior. Interestingly, despite attempts at finding concepts that are uniquely exaggerated by a specific model, we did not find any clear examples—this suggests exaggerations are approximately universal.

Overall, the examples above concretely demonstrate how conceptual blindspots manifest in generated outputs, illustrating that our energy-based diagnostic can surface both shared and model-specific failure modes. Notably, it enables the identification of surprising model limitations—such as the consistent failure to reproduce clear or solid background elements, like `whitespaces on documents`, across all models. This raises the possibility that certain failure patterns may stem from architectural constraints or training data biases that transcend individual model idiosyncrasies.

While these aggregate-level analyses are informative, they invite a deeper question: do these blindspots emerge only in the aggregate across many samples, or do they manifest themselves even at the level of individual datapoints? This finer-grained perspective allows us to probe the mechanisms behind blindspots more directly—uncovering cases of prompt misinterpretation, latent memorization, or both.

## 4.4 DATAPOINT-LEVEL ENERGY DIFFERENCE FROM INCONGRUENT TO MEMORIZED IMAGES

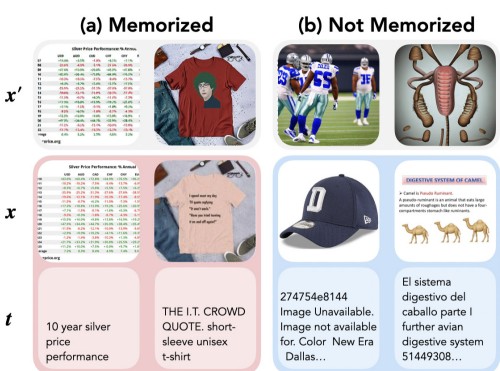

To move beyond population-level statistics, we examine individual natural vs. generated image pairs for which the $\delta(.)$ values averaged across all concepts exhibit the largest and smallest differences. This analysis aids easy understanding of model success and failures, latter of which we find often arises from prompt ambiguity or memorization artifacts. For example, Fig. 9a shows instances with near-zero difference in average $\delta(\cdot)$ values. In these cases, the generated images are conceptually indistinguishable from the original. However, qualitative inspection clearly shows this happen not because the model faithfully captures the prompt semantics, but from pure replication of memorized templates: we see repetitive visual structures (e.g., outlines of

Figure 9: ● **Datapoint-level Conceptual Alignment.** (a) Examples with minimal energy differences where models appear to memorize training patterns. (b) Examples with large differences where significant concept divergences due to prompt ambiguity or model limitations occur.

clothing or object arrangements), indicating that the model may be copying from overly frequent patterns in the training data. By contrast, Fig. 9b illustrates samples that are among the largest $\delta(\cdot)$ values. These indicate significant conceptual divergence between the synthesized and natural image. While some of these discrepancies can be attributed to underspecified or noisy captions, others reveal genuine blindspots: the prompt describes a clear concept faithfully present in $\mathcal{D}_\mathcal{X}$, yet the model fails to realize it in $\mathcal{D}'_\mathcal{X}$. This failure suggests that even when language grounding is adequate, certain concepts fall outside the model's generative abilities. To confirm that these distributional discrepancies reflect genuine failure cases rather than artifacts of poor data quality, we conduct a systematic VLM-based inspection of high-divergence samples, finding that the majority (56.3% of

the 200 most diverging datapoints) constitute genuine blindspots where the caption is sufficient but the model fails to generate the concept (see Appendix I for details).

## 4.5 Analyzing Post-Training Effects

Post-training protocols, e.g., safety fine-tuning, have been argued to reduce the diversity of model generations (Kirk et al., 2023). Given our pipeline's ability to isolate interesting differences in a model's generations and the ground-truth DGP, we next use it to understand the effects of DPO—a popular safety fine-tuning protocol (Rafailov et al., 2023). Specifically, we compare two variants of the SD 1.5 model: one trained with DPO, and one without.

For each image pair $(\mathcal{D}_{\mathcal{X}}, \mathcal{D}'_{\mathcal{X}})$, we compute the $\ell_2$ norm of the difference between their internal concept energy vectors, $\|\xi(\mathcal{D}'_{\mathcal{X}}) - \xi(\mathcal{D}_{\mathcal{X}})\|_2$. Fig. 10 presents a histogram of these datapoint-wise energy differences. The DPO-enhanced model exhibits both a lower median and a narrower spread, indicating more consistent distribution of generated concepts with the ground-truth DGP. This suggests that DPO may serve to regularize the model's concept distribution, encouraging outputs that more closely reflect the semantic content of the seen inputs. While our analysis does not disentangle the specific inductive biases introduced by DPO, these results

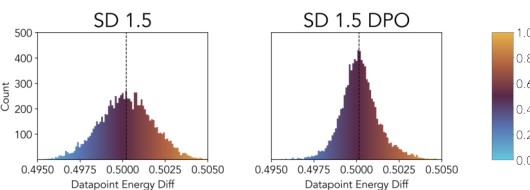

Figure 10: ● **Effect of DPO on Concept Fidelity.** Histogram of datapoint-wise energy differences between the synthesized and natural distribution of SD 1.5 models with and without DPO.

provide empirical evidence that its optimization objective, which favors human-preferred generations, indirectly promotes better match with the training distribution. In particular, it reduces *both over and under* activation of individual concepts relative to the natural baseline. These findings highlight the utility of our pipeline in characterizing the downstream effects of post-training interventions: not merely in terms of output quality, but in how they reshape the conceptual geometry of the model's output space.

## 4.6 Conceptual Misalignment as a Function of Empirical Frequency

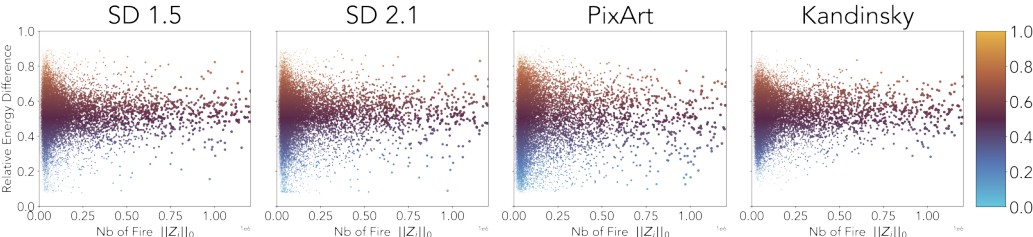

Figure 11: ● **Concept Fidelity Across Frequency Spectrum.** Scatter plots showing the relationship between concept frequency (x-axis) and the energy difference (y-axis) across four evaluated models. Each point represents a concept with size is proportional to its activation frequency.

We previously hypothesized that conceptual blindspots are not merely be architectural artifacts, but may also emerge as a direct consequence of distributional peculiarities of certain features. In this section, we empirically test this hypothesis by examining whether concepts that are rarely activated in natural images, i.e., those in the long tail of the data distribution, lead to blindspots in generative models. Specifically, we process the natural dataset $\mathcal{D}_{\mathcal{X}}$ through the trained SAE and compute, for each concept $k$, its empirical frequency $\|\boldsymbol{Z}_{:,i}\|_0$, where $\boldsymbol{Z}_{:,i}$ is the activations of concept $i$ across all our images. We then correlate this with the absolute energy difference observed across generated outputs. Fig. 11 visualizes this relationship for all evaluated models. We find that concepts with higher frequency in natural data tend to show lower energy discrepancies, while rare concepts—especially suppressed ones ($\delta(k) < 0.5$)—exhibit significant alignment errors. This suggests that many blindspots stem not from randomness or model quirks, but from systematic effects tied to long-tail concept distributions. Addressing these issues may require not just architectural changes but also strategies like data reweighting or augmentation.

## 5 Discussion

Our analysis reveals multiple conceptual blindspots in four popular generative image models. The results presented here, however, only scratch the surface: each individual finding could warrant its own dedicated investigation. Rather than delving deeply into any one of these questions, we instead showcase the versatility of our method and exploratory tool. Out of the box, they allow for a systematic identification of concepts that models struggle to generate, detection of memorization artifacts, discovery of datapoints with insufficient captions, quantification of post-training effects, and characterization of conceptual shifts across model architectures. We thus open space for follow-up work to extend the depth of analysis, scope of evaluated architectures, and inquiry into root causes of conceptual blindspots. Future work could also explore hierarchical representations of concepts in RA-SAE to allow for a more nuanced analysis. The core methodology presented in this paper is modular and agnostic to the specific concept extraction model, and such an analysis would hence require minimal adjustment to the overall process.

Beyond mere exploration and mapping of the conceptual space of existing models, our method could also serve as grounds for targeted intervention strategies employed during training of new models. Specifically, the energy profiles could inform prioritized sampling or reweighting, increasing the prevalence of suppressed concepts in the training distribution, and more. The energy profiles could also be employed into the training objective as a regularization term, explicitly penalizing deviations from the natural concept distribution.

**Limitations.** We wish to highlight several limitations of our work. By relying on DINOv2 and RA-SAE for concept extraction and representation, our approach is inherently constrained to the kinds of concepts these models capture; concepts poorly represented by them will escape our analysis. Additionally, while our sample size of 10,000 images is substantial, it may not fully capture the long tail of rare concepts, concept co-occurrence, or other compositional statistics (see Appendix K).

## Use of LLMs

Large language models (LLMs) were used in parts of the implementation and during the writing of the paper (e.g., paragraph shortening, transition refinement, etc.). AI-powered search engines were also used to help identify some references.

## Reproducibility Statement

To maximize reproducibility of our work, our code is fully open-sourced at `https://conceptual-blindspots.github.io` and the web exploratory web app is available at `https://sae-diff.github.io`. This repository will also include the Conceptual Blindspots data extracted for the models/datasets used in this paper. Furthermore, our experimental setup is clearly outlined in Appendix D.

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

APPENDIX CONTENTS

# A    RELATED WORK

## A.1    EXPLAINABILITY IN VISION

Early work in explainable AI, including computer vision, focused on methods for attribution of influential input regions Simonyan et al. (2013); Sundararajan et al. (2017); Selvaraju et al. (2017). However, these methods offered limited semantic information about learned representations and often produced incorrect explanations Adebayo et al. (2018); Ghorbani et al. (2017); Hase and Bansal (2020). To address these issues, concept-based interpretability Kim et al. (2018) emerged to identify semantically meaningful directions in neural networks, revealing not just where they look but what concepts and structures they employ Bau et al. (2017); Fel et al. (2023a); Kowal et al. (2024).

Recent work demonstrates that popular concept-based interpretability methods—ACE Ghorbani et al. (2017), CRAFT Fel et al. (2023a), and Sparse Autoencoders (SAEs)Cunningham et al. (2023); Bricken et al. (2023)—essentially address the same dictionary learning task under different constraintsFel et al. (2023b). Out of these approaches, sparse autoencoders (SAEs) have emerged as particularly scalable for concept-based interpretability. While recent studies reveal some limitations of the original SAEs—including overly specific features Chanin et al. (2024), compositionality challenges Wattenberg and Viegas (2024), and limited intervention effects Bhalla et al. (2024)—improved SAE versions have emerged, including archetypal SAE (RA-SAE) Fel et al. (2025), hierarchical approaches Bussmann et al. (2025); Zaigrajew et al. (2025), and variants addressing specific architectural choices Bussmann et al. (2024); Makhzani and Frey (2014).

Beyond SAEs, other interpretability methods include prompt-based probing Chowdhury et al. (2025), attention map or activation visualizations Bau et al. (2018); Tang et al. (2022), and dataset-level statistics Dombrowski et al. (2024); Hwang et al. (2024) (e.g., diversity or distribution coverage metrics) offer only partial insights to answer these questions. Crucially, they lack granularity, focusing on full images or prompts instead of fine-grained features and concepts Theis et al. (2015); Naeem et al. (2020). Furthermore, they depend on subjective interpretation and do not distinguish between various failure models Borji (2023). The existing methods and metrics are hence inadequate in systematically identifying feature- and concept-level weaknesses of generative image models Stein et al. (2023).

## A.2    GENERATIVE IMAGE MODELS

Diffusion-based methods have become dominant across various modalities in generative vision modeling, including image Saharia et al. (2022); Ramesh et al. (2022); Song et al. (2020a; 2023); Nichol and Dhariwal (2021), video Ho et al. (2022); Lu et al. (2023); Wang et al. (2023b); Guo et al. (2023); Lin and Yang (2024); Hong et al. (2022); Chen et al. (2023b); Wu et al. (2023b), and 3D Poole et al. (2022); Lin et al. (2023); Jun and Nichol (2023); Wang et al. (2023c). In the domain of image generation, this can be traced back to denoising diffusion probabilistic models (DDPMs) Ho et al. (2020), which were later extended to non-Markov diffusion processes with denoising diffusion implicit models (DDIMs) Song et al. (2020b).

The Stable Diffusion (SD) Rombach et al. (2022) model family made DDMPs highly accessible both in the research and open-source communities. The original SD was followed by several subsequent versions, including SD 2 Stability AI (2022), SD 3 Stability AI (2024), SD XL Podell et al. (2023). Many modifications and extensions of the SD architecture have emerged, enabling additional constraints for the diffusion process (e.g., style Sohn et al. (2023); Pan et al. (2023), pose Zhang et al. (2023), and identity Ruiz et al. (2023); Tomavsevic et al. (2025)) as well as different input modalities, such as image-to-image generation. Different Latent Diffusion Models built on top of SD—including Kandinsky Razzhigaev et al. (2023), PixArt Chen et al. (2023a), DeepFloyd Saharia et al. (2022); Stability AI (2023), and FLUX Black Forest Labs (2024)—have also emerged.

## A.3    DATASETS FOR GENERATIVE IMAGE MODELS

The recent success of generative vision models is largely attributed to the abundance of computational resources and large-scale internet datasets Dosovitskiy et al. (2020); Yu et al. (2022); Hestness et al. (2017). Specifically, LAION-5B Schuhmann et al. (2022) has played a key role in the training of open-source text-to-image models, including SD and its derivatives. This dataset, scraped from

Common Crawl Common Crawl (2017), contains over 5B image-caption pairs, 2.3B of which are in English. Other prominent datasets include COYO-700M Byeon et al. (2022) and Conceptual Captions Sharma et al. (2018), with 700M and 3M image-caption pairs, respectively.

As LAION-5B gained popularity, concerns grew over its biases Birhane et al. (2023; 2024); Seshadri et al. (2023); Birhane et al. (2021); Thiel (2023). Despite filtering attempts, harmful content persisted Birhane et al. (2023; 2024); Seshadri et al. (2023), including NSFW material Birhane et al. (2021) and hundreds of CSAM instances Thiel (2023), prompting its temporary removal from official channels. The dataset also suffers from low-quality images Shirali and Hardt (2023) and internet-style captions (e.g., product descriptions) that misalign with how users prompt trained models Nguyen et al. (2023).

### A.4 CONCEPT DISCOVERY AND SPARSE CODING IN GENERATIVE IMAGE MODELS

Dictionary learning seeks to find sparse representations of input data, where each sample can be reconstructed using a linear combination of few dictionary atoms Olshausen and Field (1996); Elad (2010); Mairal et al. (2014). Built upon compressed sensing theory Donoho (2006); Candès et al. (2006), the field evolved from early vector quantization methods Lloyd (1982) to sophisticated approaches including Non-negative Matrix Factorization Lee and Seung (1999); Gillis (2020), Sparse PCA Zou et al. (2006), and K-SVD Aharon et al. (2006). Recent advances include online methods Mairal et al. (2009), structured sparsity Jenatton et al. (2010), and theoretical guarantees Spielman et al. (2012); Barbier and Macris (2022), alongside growing connections to deep learning Papyan et al. (2017); Tamkin et al. (2023).

Advances in sparse coding have also been leveraged to study the emergence of high-level concepts inside diffusion models Tinaz et al. (2025); Surkov et al. (2024). Prior to diffusion models, concept-grounded interpretability has been deployed to earlier generative architectures through concept-bottleneck models, which require human intervention at training time Kulkarni et al. (2025), and post-hoc detectors that retrofit concept supervision Yuksekgonul et al. (2022). However, both of these approaches require human-defined concepts and hence inherently miss broader trends that the user does not explicitly register. Recent work has also explored SAE applications to understanding temporal dynamics in language models Demircan et al. (2024), hierarchical structure in vision models Olson et al. (2025), and theoretical connections between autoencoders and sparse coding through unrolled optimization approaches Gregor and LeCun (2010); Chen et al. (2018); Ablin et al. (2019); Tolooshams and Ba (2021); Malézieux et al. (2021); Arora et al. (2015); Hindupur et al. (2025).

Aggregate metrics (e.g., precision, recall, density, and coverage Kynkäänniemi et al. (2019); Naeem et al. (2020)) and latent density scores, which predict sample quality based on the model's latent space Xu et al. (2024), have emerged to evaluate generative image model capabilities. While these effectively uncover distributional gaps, they offer little insight into specific concepts that are under- or over-represented.

Together, these limitations motivate a scalable, unsupervised framework that can systematically identify and quantify concept-level failure modes in generative image models Laina et al. (2022).

# B   COMPARISON WITH EXISTING APPROACHES

Table 1 summarizes the existing approaches for evaluating consistency and semantic coverage of generative image models, and compares them to our method.

**Fréchet Inception Distance (FID), Heusel et al. (2017).**   This metric embeds images using Inception-v3 and calculates the Wasserstein-2 distance between natural and generated distributions. While it is regularly used to encode the overall quality of a model, it aggregates many potential failure modes into a single scalar. It is hence incapable of surfacing specific conceptual blindspots.

**CLIPScore, Radford et al. (2021).**   CLIPScore computes the cosine similarity between the embeddings of a text prompt and a generated image to assess their consistency. This method is bound by the prompt itself; it cannot detect blindspots for concepts that are not explicitly included in the evaluation prompt set.

**Improved Precision and Recall, Kynkäänniemi et al. (2019).**   This evaluation framework estimates the manifold of real and generated data using k-Nearest Neighbor (k-NN) radii to separately quantify precision (fidelity) and recall (coverage). While a drop in recall implies the existence of distributional blindspots (mode collapse), the metric cannot identify *which* concepts are missing.

**GAN Dissection, Bau et al. (2018).**   This method correlates the activation maps with semantic segmentation masks to identify units responsible for specific concepts. Although it offers high granularity, it is computationally intensive and limited to the fixed vocabulary of the external segmentation network used for supervision.

**Adversarial Search (SAGE), Liu et al. (2023a).**   SAGE treats the generative image model as an adversary and optimizes over text tokens to discover prompts that maximize divergence from a surrogate classifier. While effective at identifying specific error cases, it lacks a structured representation of the full conceptual space, and the iterative optimization makes it prohibitively expensive.

**Concept Bottleneck Models (CBMs), Koh et al. (2020).**   These architectures explicitly force the neural network to compress information into a layer where neurons correspond to pre-defined human concepts. This requires training models from scratch with concept-labeled data, making it unsuitable for the post-hoc evaluation of pre-trained foundation models.

Table 1: Comparison of Approaches to Identifying Conceptual Blindspots in Image Models.

| Method | Specificity | Unsupervised | Scalability | Exaggeration |
|---|---|---|---|---|
| FID | ✗ | ✓ | ∼ | ✗ |
| Precision & Recall | ✗ | ✓ | ∼ | ✗ |
| CLIPScore | ✗ | ✗ | ∼ | ✗ |
| GAN Dissection | ✓ | ✗ | ✗ | ✓ |
| SAGE | ✓ | ✗ | ✗ | ✗ |
| CBMs | ✓ | ✗ | ✗ | ✗ |
| Human Evaluation | ✓ | ✓ | ✗ | ✓ |
| **Our Method** | ✓ | ✓ | ✓ | ✓ |

**Evaluation Criteria**

1. **Specificity.** The method supports a notion of discrete concepts (these may be defined in text, via examples, a dictionary or otherwise; there may also exist a taxonomy/hierarchy).
2. **Unsupervised.** For a specific concept to be deemed as a blindspot, the method does not require for the user to explicitly define or describe it.
3. **Scalability.** The complete conceptual space of the evaluated model, as conceptualized by the framework, can be feasibly searched (given a conceptual space of $\geq 1,000$ concepts).
4. **Exaggeration.** The method can detect *both* suppression and exaggeration.

## C    EXPLORATORY TOOL

Shown in Figure 12 is an overview of the exploratory tool developed alongside this project to facilitate inspection and comparison of concept-level energy differences. The tool is a web-based interface built around a UMAP projection of concept representations, enabling visualization and comparison of concept-level energy differences. It is publicly available at `https://sae-diff.github.io/`, along with pre-computed energy difference data for the four models evaluated in this work (SD 1.5, SD 2.1, PixArt, and Kandinsky). All subsequent analyses in this paper are derived from insights enabled by this tool. Its primary functionalities, which support these analyses, include:

- **Contrast different models and architectures.** For each evaluated model, the tool provides a UMAP visualization spanning all 32,000 concepts from the RA-SAE. Each scatter point represents an individual concept, color-coded by its energy difference.
- **Inspect concepts.** Each concept has a card with key statistics, representative real and generated images ($x$, $x'$), and visualized co-occurrence patterns.
- **Explore blindspots.** Beyond the UMAP and per-concept views, the tool features global rankings of suppressed and exaggerated blindspots, helping to highlight the most notable conceptual blindspots.

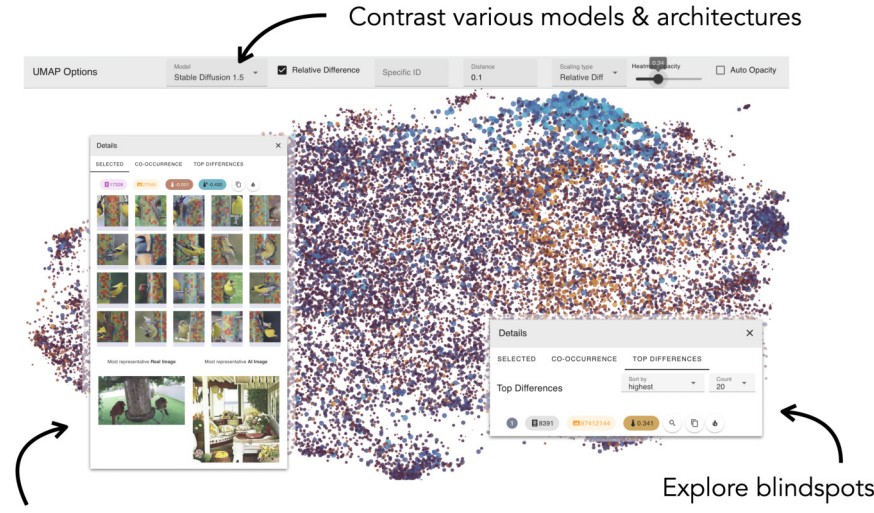

Figure 12: **Overview of the Exploratory Tool.** The web interface displays a UMAP projection for each evaluated model, where each dot represents a concept, color-coded by its energy difference. When a concept is selected, a detail panel presents illustrative images, statistics, and the most representative natural and generated images $x$ and $x'$. An ordered list of the concept's co-occurrences is shown alongside global rankings of blindspots.

## D    EXPERIMENTAL SETUP

This section details the experimental setup for our analysis of four popular generative image models: SD 1.5/2.1, Kandinsky, and PixArt, all trained on LAION-5B or its subsets/derivatives. The code is available at `https://github.com/sae-diff/code-review`.

### D.1    OBSERVATION SPACE

The observation space is constructed by sampling $10,000$ image-text pairs from the LAION-5B dataset Schuhmann et al. (2022), which serves as our domain of natural images. Due to concerns with CSAM and other unsafe content in the dataset, the original data release is no longer available. A substitute release of a subset of this dataset with additional filtering of the unsafe content, available at `https://huggingface.co/datasets/laion/relaion2B-en-research-safe`, is used.

The sampling procedure consists of: (1) loading the full LAION dataset using the Hugging Face `datasets` library, (2) performing validation to ensure proper URL structure and resource availability via HTTP HEAD requests, and (3) employing random sampling with replacement until reaching the target count of $10,000$ valid samples. This approach, yielding $D_G$ with $(x, t)$ tuples, ensures our observation space contains accessible image-text pairs for comparative analysis of a dataset of image URLs whose large portion has been made unavailable since original release. Additional examples of synthesized images are shown in Appendix N.

### D.2    SYNTHESIZED IMAGES

For each of the four evaluated models, we generate a synthetic dataset $D_{g_\theta}$ to have a one-to-one correspondence with $D_G$, yielding triplets $(x, x', t)$. Specifically, given the 10,000 image-text pairs $(x, t)$ from $D_G$, we use $t$ to synthesize counterpart images $x'$ using each generative model $g_\theta$.

The synthesis process follows the standard text-to-image generation pipeline for each model architecture, implemented using the Hugging Face `diffusers` library, where the models are loaded at mixed precision (`fp16`). All synthetic images are generated at $512 \times 512$ pixel resolution with default parameters.

**Stable Diffusion 1.5.**  The checkpoint from `https://huggingface.co/benjamin-paine/stable-diffusion-v1-5` (which is a mirror of the deprecated `https://huggingface.co/ruwnayml/stable-diffusion-v1-5`) is used. Inference is performed using $50$ inference steps, with the guidance scale fixed at $7.5$.

**Stable Diffusion 1.5 + DPO.**  The DPO variant of SD 1.5 (used in the analysis in Sec. 4.5) follows the baseline SD 1.5 implementation, but replaces the UNet component with a DPO-trained version from `https://huggingface.co/mhdang/dpo-sd1.5-text2image-v1`.

**Stable Diffusion 2.1.**  The checkpoint from `https://huggingface.co/stabilityai/stable-diffusion-2-1` is used. Inference is performed using $50$ inference steps, with the guidance scale fixed at $7.5$.

**Kandinsky.**  The checkpoint from `https://huggingface.co/kandinsky-community/kandinsky-2-1` is used. Inference is performed using $100$ inference steps, with the guidance scale fixed at $4.0$.

**PixArt.**  The checkpoint from `https://huggingface.co/PixArt-alpha/PixArt-XL-2-1024-MS` is used. Inference is performed using $50$ inference steps, with the guidance scale fixed at $7.5$.

### D.3    ● DISTRIBUTION LEVEL ANALYSIS

**Section 4.1.**  We compute energy differences $\delta(\cdot)$ across all 32,000 concepts for each evaluated model. The sigmoid transformation with temperature $T = 0.8$ is applied during normalization. The resulting values are visualized as log-scale density histograms with 100 bins spanning $[0, 1]$.

**Section 4.2.** We embed the complete set of 32,000 concepts into two-dimensional space using UMAP applied to the sparse concept codes. Each point in this UMAP represents an individual concept, colored according to its energy difference $\delta(.)$, emphasizing both suppressed and exaggerated blindspots. To quantify cross-model consistency, we compute pairwise Pearson correlation coefficients between $\delta(.)$ vectors of all model pairs, producing both scatter plots and correlation matrices. This analysis reveals whether blindspots cluster in conceptual space and identifies model-specific versus universal patterns of conceptual blindspots.

**Section 4.3.** We rank all 32,000 concepts by their energy difference $\delta(.)$, and manually examine the extrema (both suppressed and exaggerated blindspots). For suppressed blindspots, we select concepts with $\delta(.) < 0.1$; for exaggerated blindspots, we choose those with $\delta(.) < 0.9$. Presented examples are manually annotated with textual descriptions of the respective concepts through inspection of their most activating images and spatial attention patterns. We outline ongoing efforts to automate this concept interpretation in Appendix F.2.

**Section 4.5.** We compare 1.5 with and without DPO in the following fashion: for each image pair $(\boldsymbol{x}, \boldsymbol{x}')$, we compute the L2 norm of the difference between their concept energy vectors $\|\xi(\boldsymbol{x}') - \xi(\boldsymbol{x})\|_2$. We apply a sigmoid transformation with temperature $T = 0.8$ to the element-wise differences before taking their mean. This yields datapoint-wise energy differences that quantify how much each generated image deviates from its natural counterpart in concept space. Finally, these differences are visualized as overlapping histograms, contrasting both model variants.

**Section 4.6.** For each concept $c_k$, its empirical frequency $\|\boldsymbol{Z}_{:,i}\|_0$ (the count of non-zero activations across the natural dataset) is counted. A sigmoid normalization with temperature $T = 0.4$ is then applied to the energy differences $\delta(.)$ The analysis is visualized using scatter plots where the x-axis is the empirical concept frequency and the y-axis is the sigmoid-transformed energy difference. The point sizes are proportional to activation frequency and point colors are proportional to the magnitude of energy differences.

### D.4 ● DATAPOINT LEVEL ANALYSIS

**Section 4.4.** For each image pair $(\boldsymbol{x}, \boldsymbol{x}')$, we compute the L2 norm of the difference between their concept energy vectors $\|\xi(\boldsymbol{x}') - \xi(\boldsymbol{x})\|_2$. This yields a scalar measure of conceptual divergence for each image pair. The samples are ranked by their energy differences. Minimal divergence indicate potential memorization artifacts and maximal divergence point to significant conceptual failures. This analysis enables qualitative inspection of specific failure modes.

### D.5 ● CO-OCCURRENCE ANALYSIS

**Appendix H** For both the natural and synthesized data $D_{\boldsymbol{G}}$ and $D_{\boldsymbol{g_\theta}}$, concept co-occurrence patterns are analyzed through the co-activation matrix $Z^T Z$, which holds pair-wise correlations in concept usage. Spectral analysis is performed to examine the dominant conceptual directions using eigendecomposition. The alignment between natural and synthetic co-occurrence structures is assessed using cosine similarity heatmaps between the top-100 eigenvectors of each co-occurrence matrix. These $100 \times 100$ similarity matrices are visualized as square heatmaps where perfect diagonal alignment would indicate identical principal concept axes, while off-diagonal patterns reveal would revolve rotations and mismatches in compositional geometry.

# E    COMPUTATIONAL RESOURCES

This section summarizes the GPU resources used for training and experiments in support of this paper. In total, we used approximately 202 GPU-hours on NVIDIA H100s and H200s.

**RA-SAE.**    Trained for approximately 24 GPU-hours on three NVIDIA H100s.

**Synthesized Images.**    Generating the full $D_{g_\theta}$ (see Appendix D) took roughly 5 hours per generator when distributed across four NVIDIA H200 GPUs. With five generators, this totaled approximately 100 GPU-hours on a NVIDIA H200.

● **Distribution-Level Analysis.**    Extracting energy differences at the distribution level took about 3 hours per generator on a single NVIDIA H200 GPU (total $\sim 15$ GPU-hours).

● **Datapoint-Level Analysis.**    Computing datapoint-level energy differences, ranking concepts per datapoint, and ranking datapoints per concept also took approximately 3 hours per generator on one H200 GPU (total $\sim 15$ GPU-hours).

## F  CUSTOM RA-SAE

This section introduces our custom relaxed archetypal sparse autoencoder (RA-SAE), its training configuration, autointerpretability pipeline, and examples of learned concepts. The model is open-sourced at `anonymized`.

**Sparse Autoencoders.**  SAEs decompose the high-dimensional activation space of models into sparse, human-interpretable concepts. Specifically, SAEs enforce a sparsity constraint so that each activation vector is reconstructed using only a small subset of learned feature directions (i.e. concepts), which helps disentangle overlapping information (superposition) into more separable parts. Once trained, each concept is assigned a human-interpretable label, either by human annotators or via autointerpretability pipelines (for example using a vision-language model). To assist with this labeling, high-activating exemplars (inputs that yield strong activation for that concept) are identified, and recurring visual or semantic patterns across those exemplars are described.

**Archetypal SAEs.**  Regular SAEs suffer from instability: small changes in initialization, data, or training can lead to different learned dictionaries. Archetypal SAEs (A-SAEs) mitigate this by constraining dictionary atoms (feature directions) to lie within the convex hull of the data; that is, each concept vector must be expressible as a convex combination of actual activation vectors from the data. This geometric anchoring forces the learned features (atoms) to be more directly tied to the underlying data, improving stability. Relaxed Archetypal SAEs (RA-SAEs) loosen this constraint somewhat to allow more flexibility (better reconstruction ability) while retaining much of the stability benefits. Empirically, RA-SAEs have been found to match or outperform regular SAEs in benchmarks of plausibility (how well learned directions recover known classification or semantic directions) and identifiability (how well they disentangle synthetic mixtures of concepts), producing more stable and semantically meaningful concepts.

**Our Configuration.**  Our custom RA-SAE was trained on top of DINOv2 representations using the Top-K sparsity constraint Gao et al. (2025). It has $32,000$ concepts, making it largest RA-SAE to date. Training details are reported in App. F.1. We describe the autointerpretability pipeline to assign labels to concepts in App. F.2. Finally, examples of concepts learned by the RA-SAE are given in App. F.3

### F.1  TRAINING DETAILS

**Dataset.**  The auto-encoder is trained on the complete ImageNet-1k training split, ($\approx 1.28\,\mathrm{M}$) RGB images. Each image is converted to 261 visual tokens using DINOv2 Oquab et al. (2023); tokens are fed to the SAE without class or position embeddings. The total number of training tokens is therefore $50 \times 1.28\mathrm{M} \times 261 \approx 1.67 \times 10^{10}$.

**Dictionary.**  The dictionary has $32{,}000$ concept dimensions. For the sparse activation rule, top-$k$ masking with $k = 5$, is used; activations outside the largest five per input are set to 0. The weights are initialized using Xavier/Glorot. The training is conducted at mixed precision (`fp16`), with the last ten epochs performed at full precision.

**Optimizer and Schedule.**  The model is trained for 50 epochs using base AdamW ($\beta_1 = 0.9$, $\beta_2 = 0.999$) optimizer is employed with weight decay set to $10^{-5}$. Linear warm-up is applied on the first $5\%$ of steps, followed by cosine decay from $\eta_{\mathrm{max}} = 5 \times 10^{-4}$ to $\eta_{\mathrm{final}} = 10^{-6}$. MSE loss is used alongside an auxiliary term penalizing activations that never enter the top-$k$ set, where $\lambda = 10^{-5}$.

### F.2 AUTOINTERPRETABILITY

In an effort to automate the interpretation of concepts identified by the RA-SAE, we qualitatively evaluated the ability of Vision-Language Models (VLMs) to describe them. In particular, we queried ChatGPT 4o via the OpenAI API using various prompts and formats of an image $x''$ in which the given concept was salient, recording the description generated by the VLM. In this section, we highlight the findings from our exploration. For evaluation purposes, we allowed the VLM to provide its full reasoning; to make this a scalable solution, a simple modification of the prompt—asking the VLM to provide only the description—would filter out the reasoning.

First, we provided the VLM with two images: a raw version of $x''$ and $x''$ with a heatmap superimposed to localize the intensity of the concept:

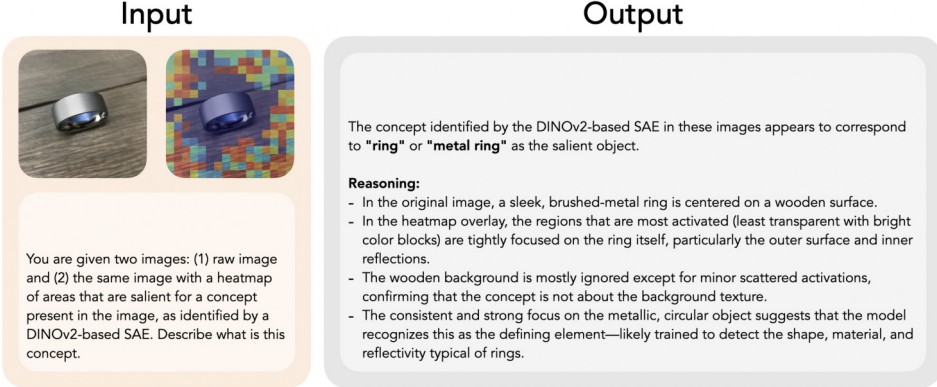

As seen in the example, the VLM did not interpret the heatmap visualization correctly. Instead of describing the wood texture in the background—the actual salient region—it described the ring in the foreground, which received no attention from the RA-SAE.

To determine whether this was a prompting issue, we tested the same scenario again—this time using an expanded prompt that explained the heatmap in greater detail:

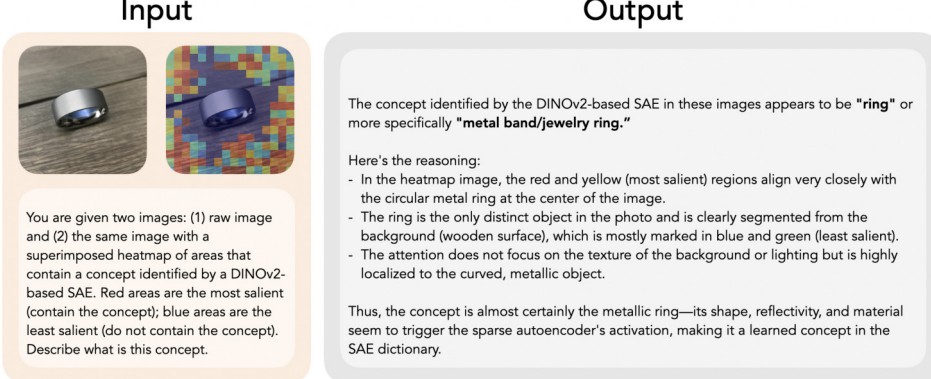

This prompt modification did not resolve the issue, as the VLM still identified the ring as the prominent concept in the image. Several other prompting techniques, including those describing the heatmap in even greater depth (e.g., by including the code used for its generation), were explored with similar results.

We next tested providing the VLM only with the heatmap-superimposed $x''$ and a similar prompt explaining how to interpret the heatmap:

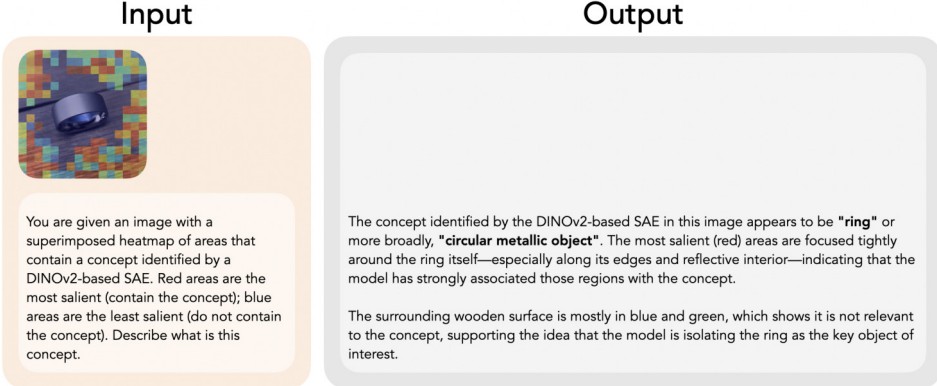

Once again, the result failed to identify the wooden background as the prominent concept in the image and instead focused on the ring. As in the previous case, we experimented with various prompting strategies using this image format, but the results remained largely unchanged.

From these observations, we concluded that the heatmap alone is insufficient to guide the VLM's attention to the salient areas. The presence of less relevant regions appears to distract the model. To address this, we replaced the heatmap-superimposed $x''$ with an alpha-masked version of $x''$:

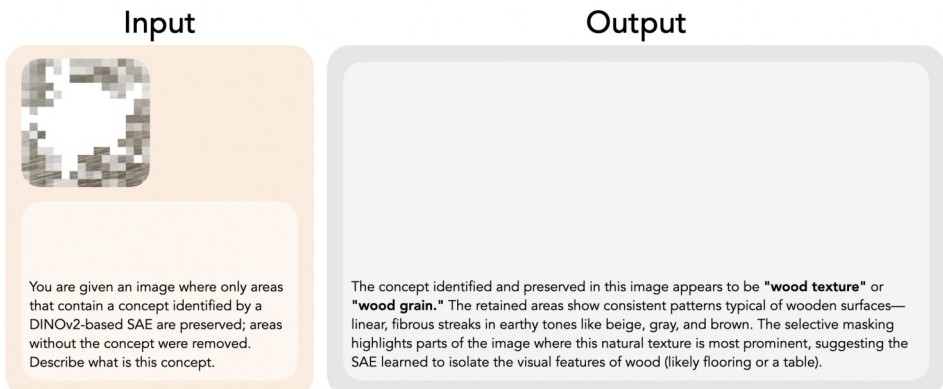

As shown in the example, the VLM was now able to correctly identify wood texture as the dominant concept. We observed consistent success across a range of concepts and images.

The formatting of $x''$ and the prompt shown above yielded the most reliable results in our qualitative evaluation. However, we note that this evaluation is limited by its qualitative nature (due to the absence of ground truth annotations) and its focus on a single VLM. We hope future work on the autointerpretability of SAE concepts can build on and expand this analysis.

### F.3 Examples of Learned Concepts

Shown below is a representative sample of concepts learned by our RA-SAE. For each concept, we present 12 images for which the concept had the highest activation in the ImageNet dataset (on the left) and a localization of the respective concept within those images (on the right). Additionally, an epitome constructed using the *Feature Accentuation* method from Hamblin et al. (2024) is shown bottom left.

As can be seen from this sample, the granularity of concepts varies. We see concepts for objects (e.g., `colorful underwater fish`), textures and patterns (e.g., `colorful polka dots pattern`), composition (e.g., `person on the right edge at social gatherings` and `bright colorful backgrounds`), actions (e.g., `skiing action on snowy slopes` and `gripping various tools and objects`), types of images (e.g., `comic book illustrations and characters` and `male portraits in various attire`), and more.

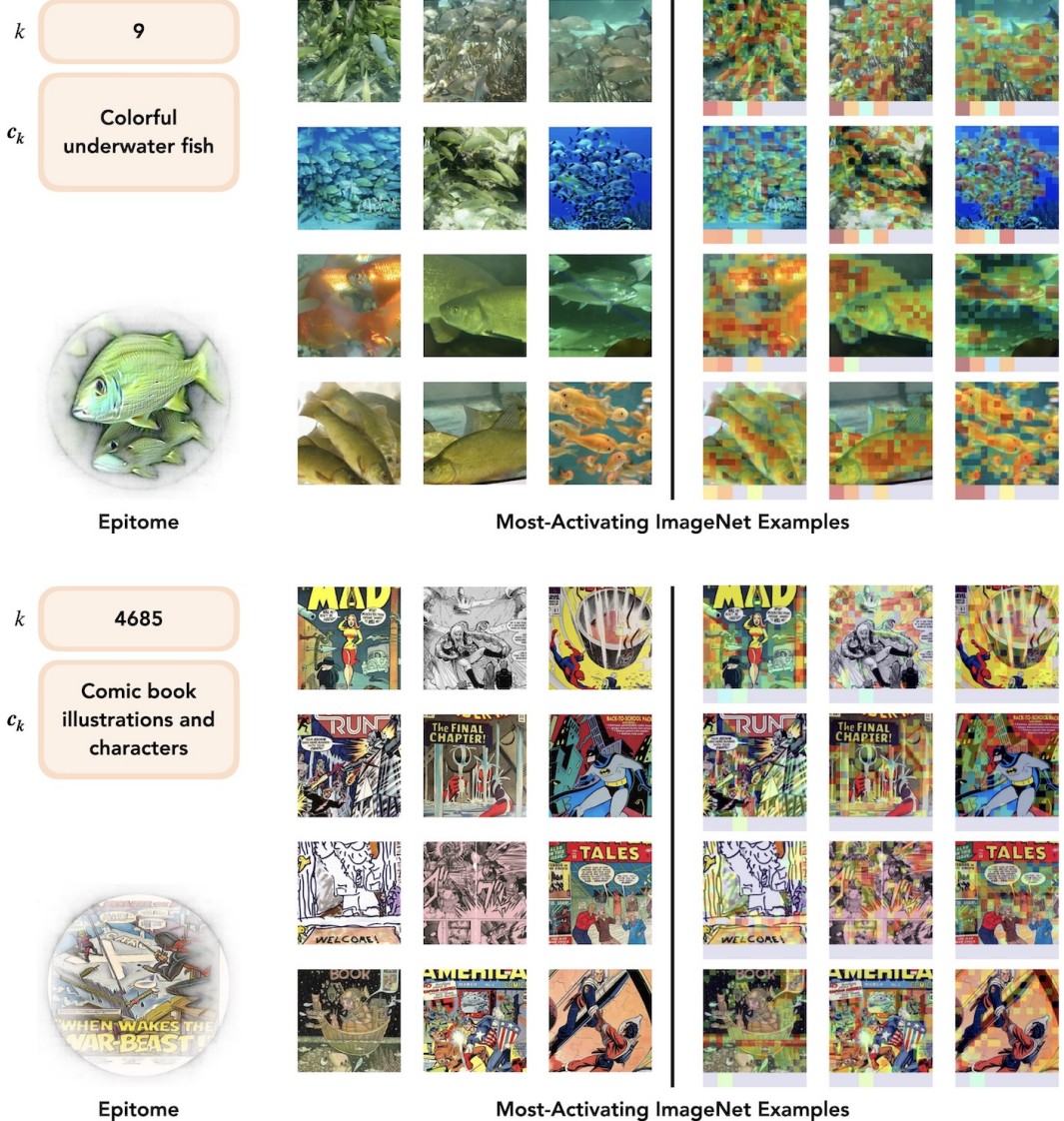

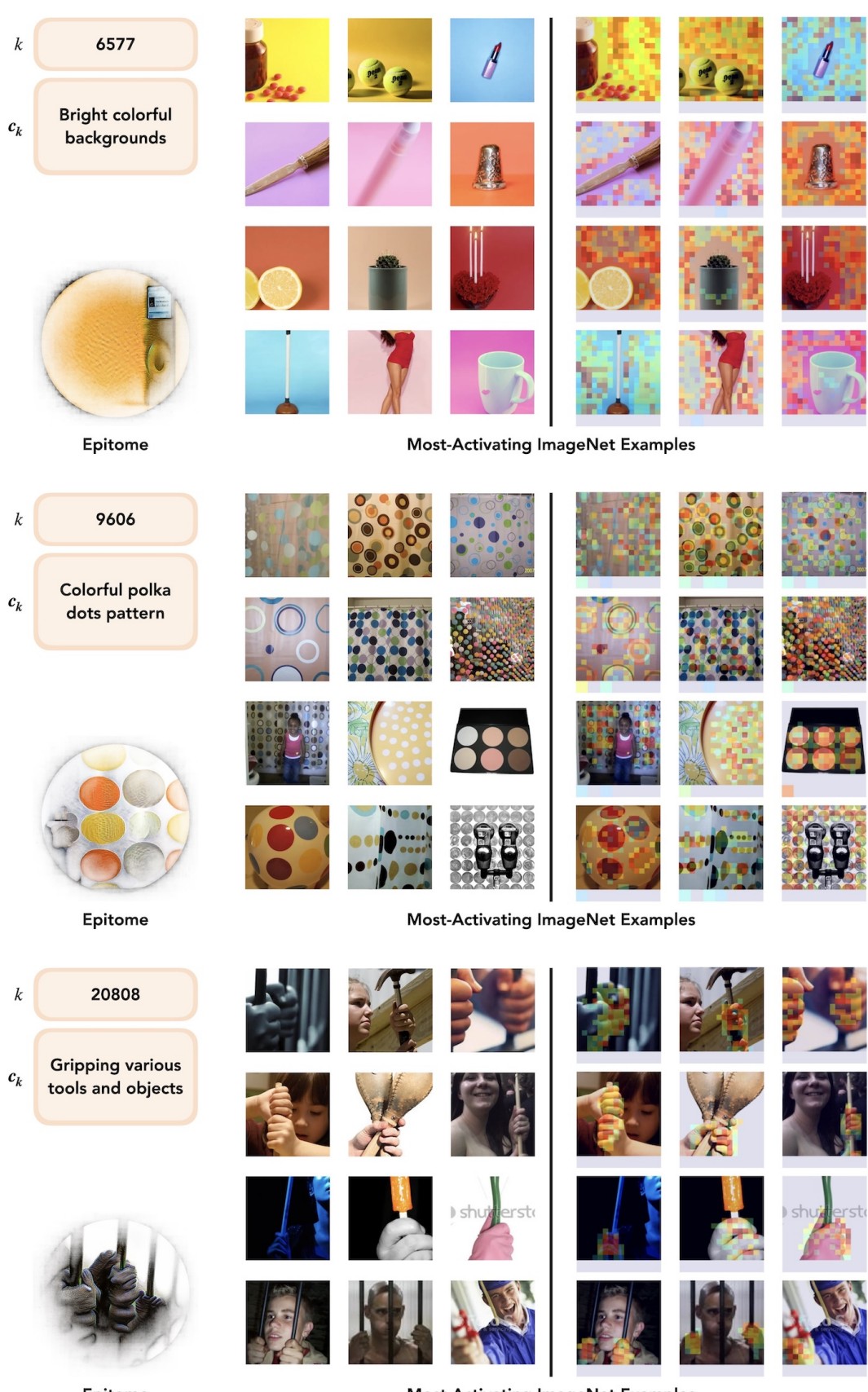

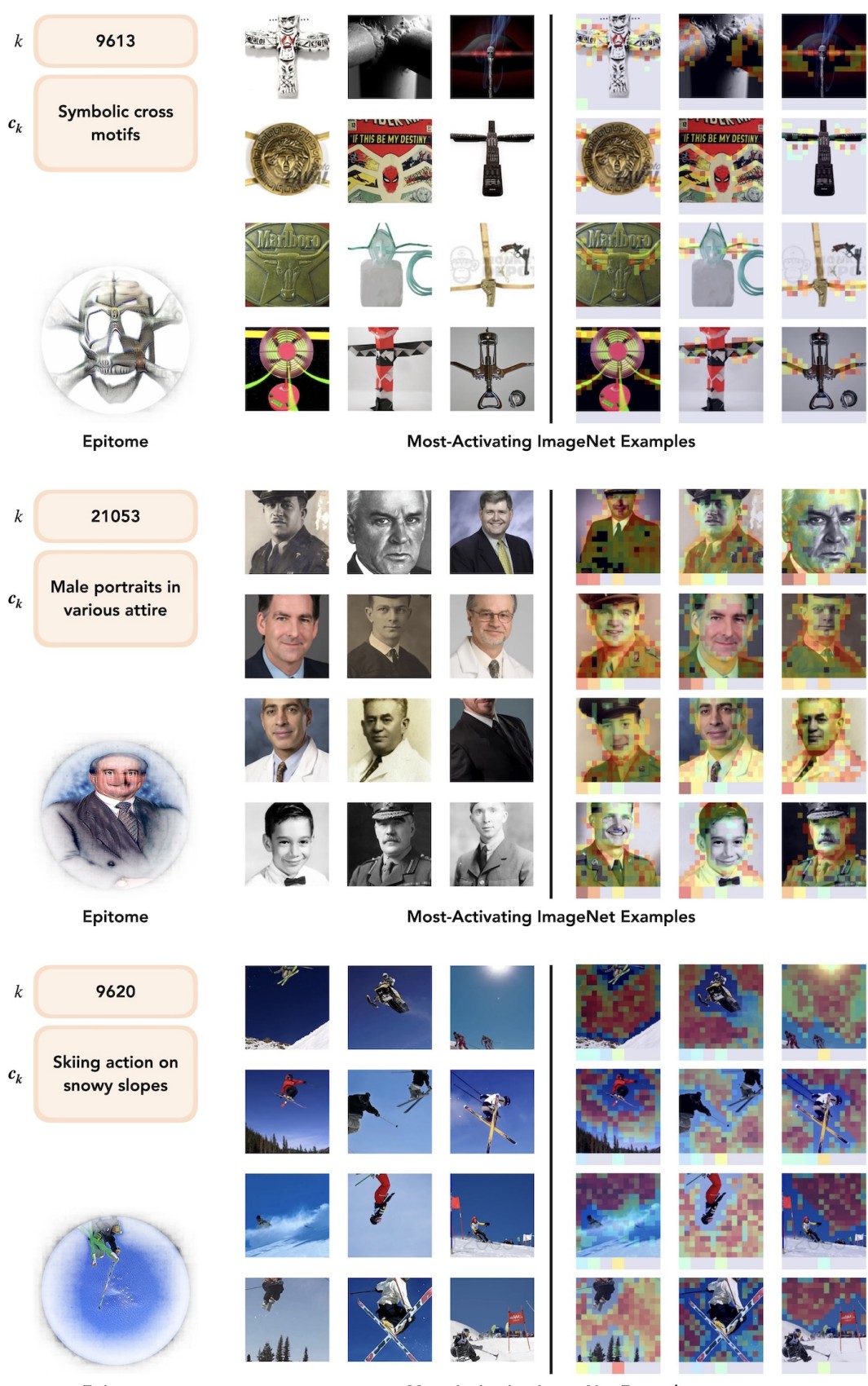

k      9613

$c_k$      Symbolic cross motifs

Epitome          Most-Activating ImageNet Examples

k      21053

$c_k$      Male portraits in various attire

Epitome          Most-Activating ImageNet Examples

k      9620

$c_k$      Skiing action on snowy slopes

Epitome          Most-Activating ImageNet Examples

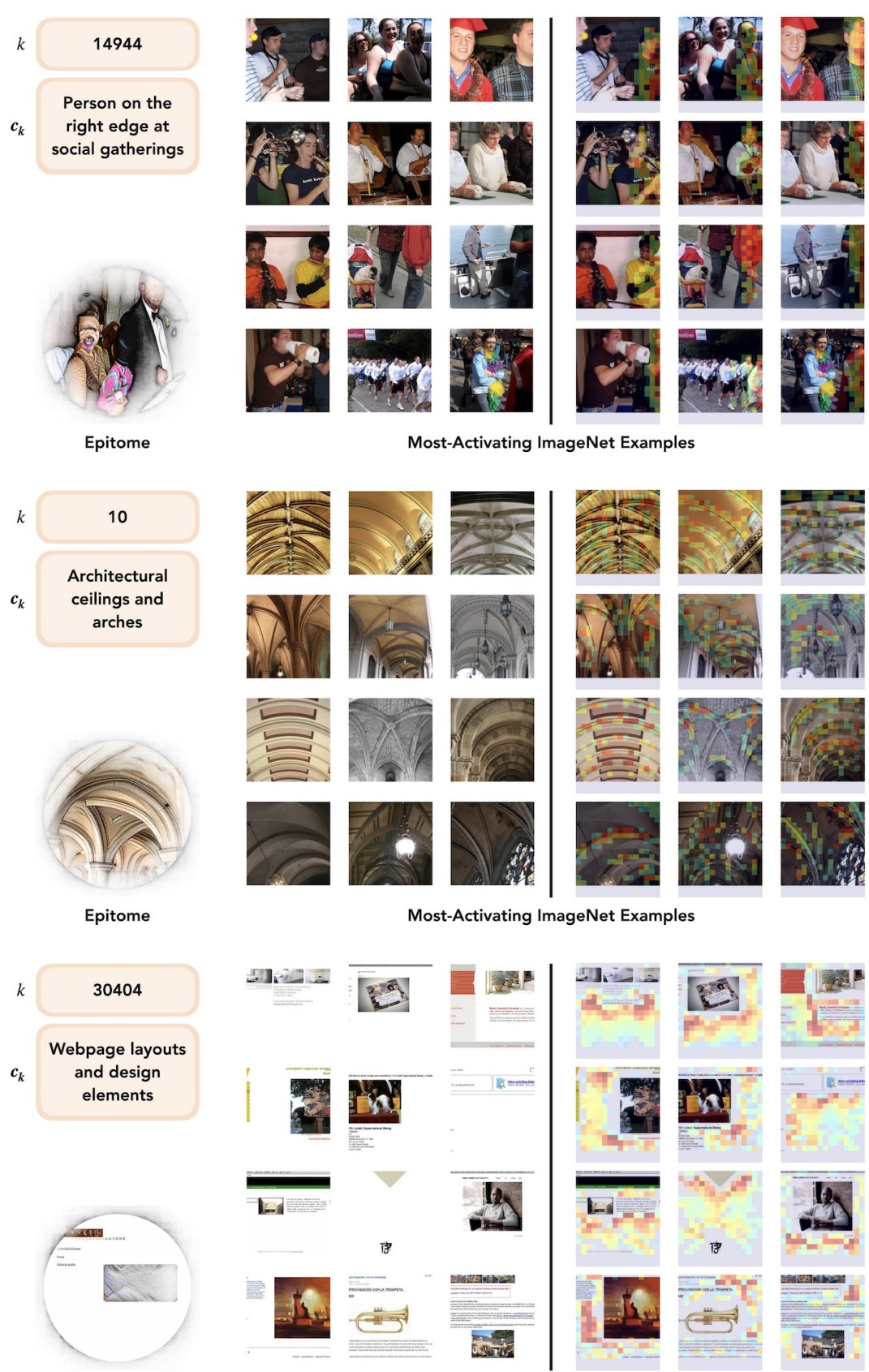

# G    ADDITIONAL RESULTS: QUALITATIVE EXAMPLES OF BLINDSPOTS

Shown below are qualitative examples of suppressed conceptual blindspots in SD 1.5. For each concept, we include a representative image from the natural distribution to illustrate the desired visual depiction. To the right, we show four images generated by SD 1.5 using various prompts designed to elicit the concept. Despite using simple, clearly worded prompts, the model consistently struggles to generate these concepts, supporting their identification as suppressed conceptual blindspots.

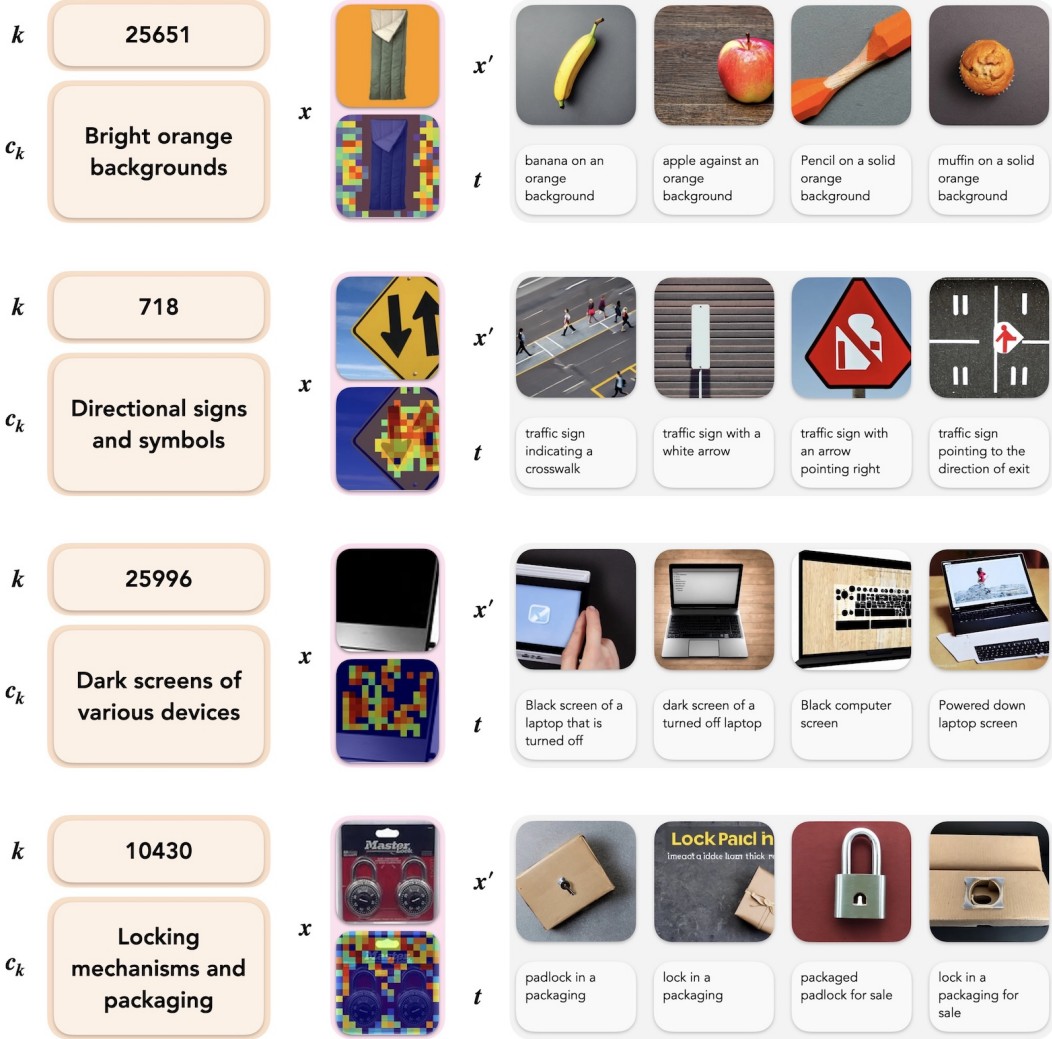

In the remainder of this section, we enumerate additional suppressed and exaggerated blindspots for each evaluated model (see App. G.1). We also describe our stress-testing procedure, in which we attempt to elicit the concepts identified as suppressed blindspots using many prompt variations, in order to validate that these are indeed true blindspots (see App. G.2).

### G.1 MODEL-SPECIFIC BLINDSPOTS

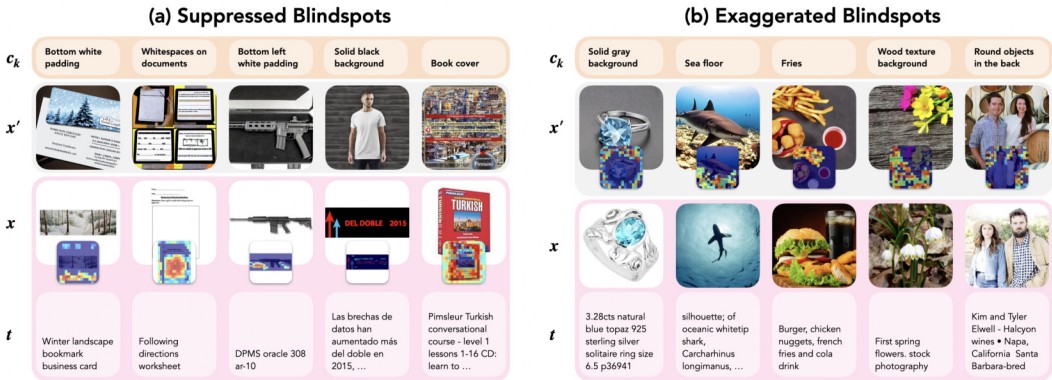

Figure 13: Examples of conceptual blindspots in **Stable Diffusion 1.5**. For each concept, the prototypical natural (for suppressed blindspots) or synthesized (for exaggerated blindspots), based on the highest absolute activation, is shown. The spatial heatmap for the concept is superimposed atop the image.

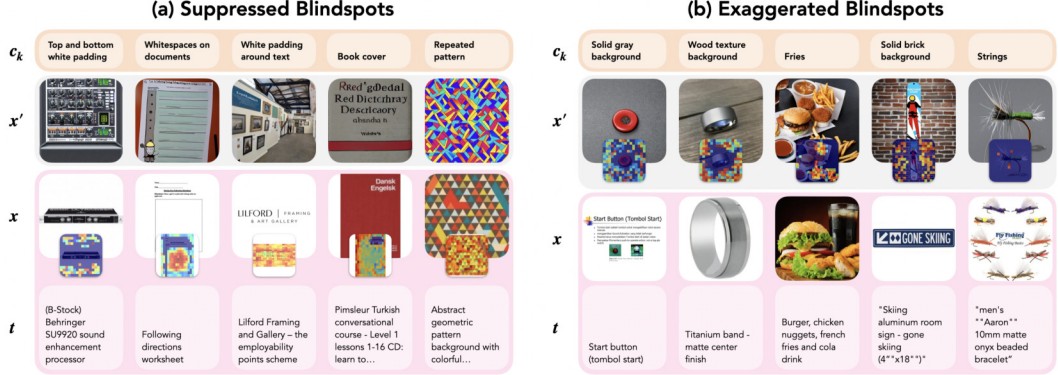

Figure 14: Examples of conceptual blindspots in **Stable Diffusion 2.1**. For each concept, the prototypical natural (for suppressed blindspots) or synthesized (for exaggerated blindspots), based on the highest absolute activation, is shown. The spatial heatmap for the concept is superimposed atop the image.

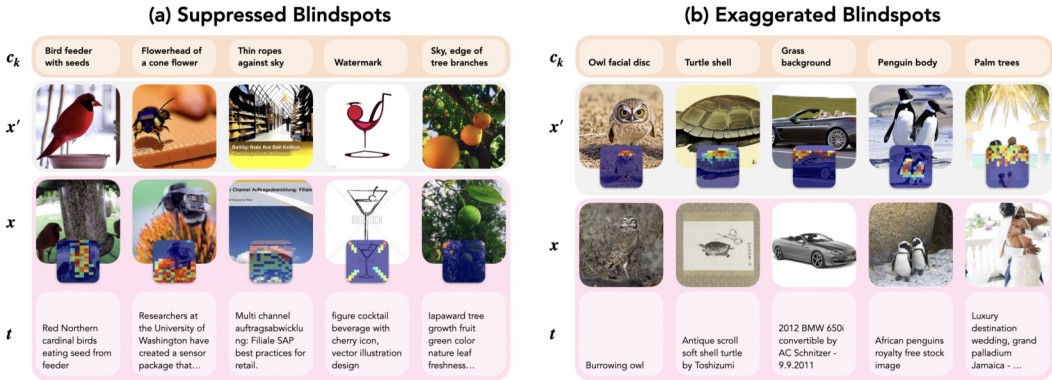

Figure 15: Examples of conceptual blindspots in **Kandinsky**. For each concept, the prototypical natural (for suppressed blindspots) or synthesized (for exaggerated blindspots), based on the highest absolute activation, is shown. The spatial heatmap for the concept is superimposed atop the image.

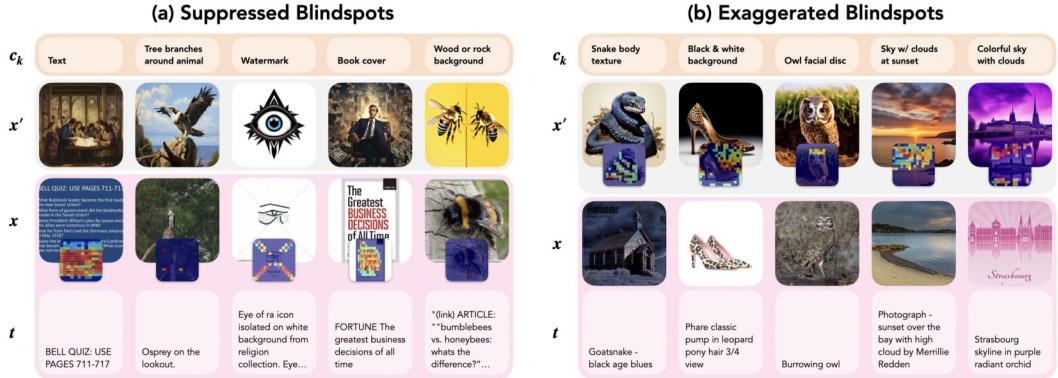

Figure 16: Examples of conceptual blindspots in **PixArt**. For each concept, the prototypical natural (for suppressed blindspots) or synthesized (for exaggerated blindspots), based on the highest absolute activation, is shown. The spatial heatmap for the concept is superimposed atop the image.

## G.2 STRESS TESTING

To stress-test the blindspots identified by our method, we gathered a range of prompts describing these blindspots and used them to generate many images. We then contrasted the outputs from models in which the concept was identified as a blindspot with those in which it was not.

Specifically, ChatGPT-4o was prompted as follows: *I want to generate an image of the following concept: "<blindspot>". Suggest 50 prompts highlighting this concept to be used as input for a text-to-image model. Return these as a list of strings in Python.* Five images were generated per prompt and analyzed using our custom RA-SAE model (see Appendix F.1), which ranked them by the intensity with which the desired concept appeared. All images were then manually reviewed to determine whether the blindspot was successfully depicted.

As seen in the following examples of suppressed concepts, while some aspects of the target concept occasionally appeared (e.g., a holder or string for the `bird feeder` blindspot and a round hole for the `glossy DVD disc` blindspot), the models generally failed to generate the full concept. This aligns with our method's assessment and supports the validity of the stress test.

### G.2.1 `Bird Feeder` BLINDSPOT IN KANDINSKY

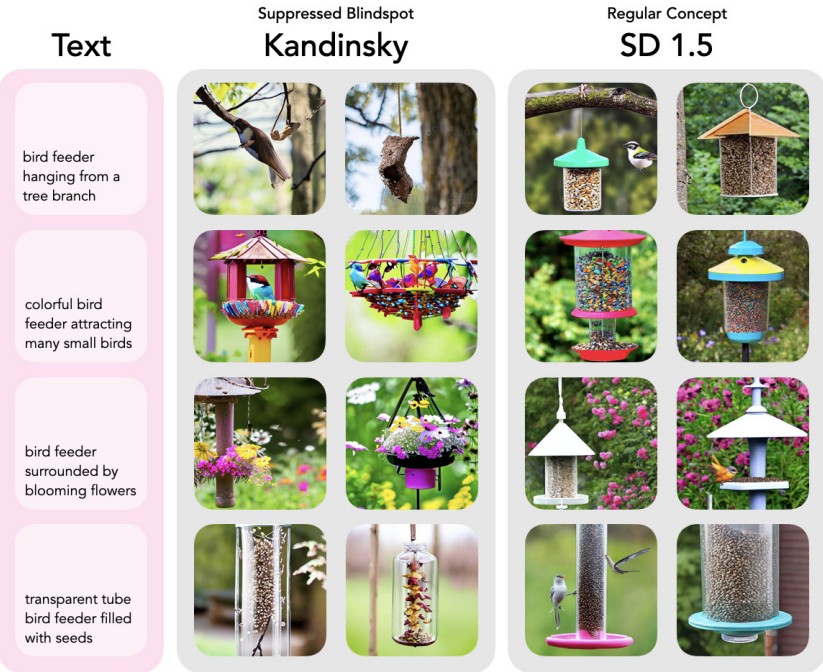

Figure 17: Examples of images generated with various prompts involving the `bird feeder` concept as a part of the stress testing. In Kandinsky, our method identified this concept as a suppressed conceptual blindspot, which matches the observed behavior: the model is unable to generate a corresponding image. By contrast, SD 1.5, in which this concept was not identified as a blindspot, is able to generate this concept.

### G.2.2   `Glossy DVD Disc` BLINDSPOT IN SD 1.5

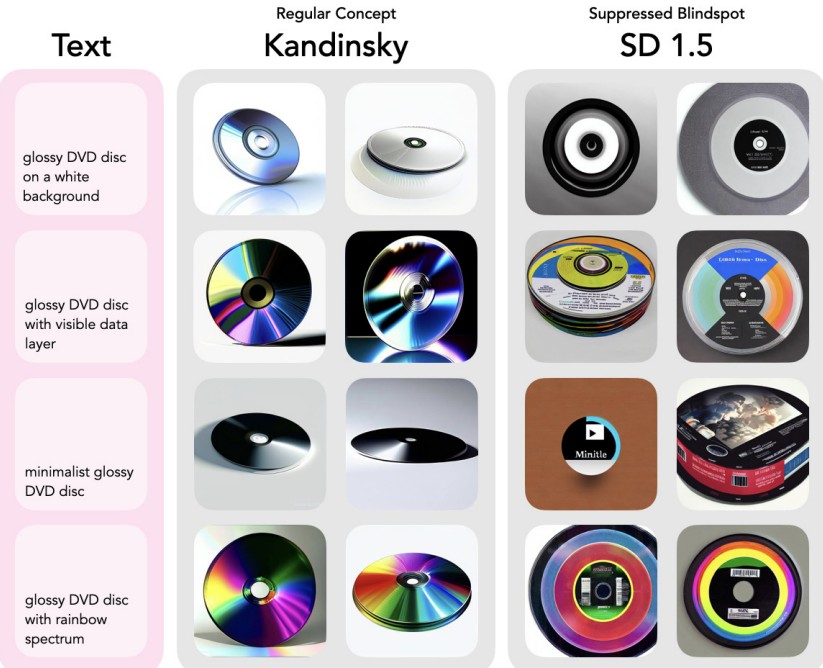

Figure 18: Examples of images generated with various prompts involving the `glossy DVD disc` concept as a part of the stress testing. In SD 1.5, our method identified this concept as a suppressed conceptual blindspot, which matches the observed behavior: the model is unable to generate a corresponding image. By contrast, Kandinsky, in which this concept was not identified as a blindspot, is able to generate this concept.

## H ADDITIONAL RESULTS: HIGHER-ORDER BLINDSPOTS WITH COMPOSITIONAL DISCREPANCY

Thus far, our analysis has centered on individual concept activations. Yet visual scenes are rarely composed of isolated concepts; instead, they are structured through rich and structured co-occurrence patterns that encode compositional semantics. We now examine whether generative models capture this higher-order structure by analyzing the co-activation matrix $\boldsymbol{Z}^{\mathsf{T}}\boldsymbol{Z}$, which reflects pairwise correlations in concept usage.

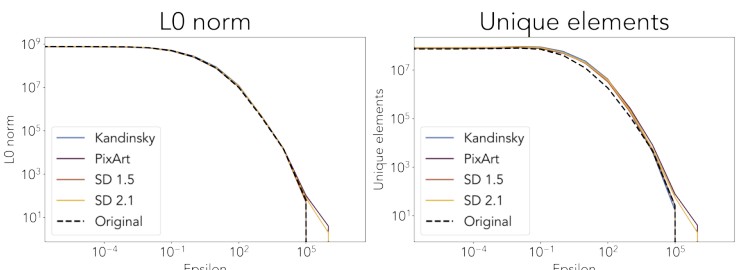

Figure 19: ● **Sparsity and Structural Divergence.** On the left: L0 norm of the co-occurrence matrix $ZZ^T$ as a function of $\epsilon$ (threshold), indicating how many entries remain active in each model. On the right: Number of unique entries in the synthesized distribution relative to the natural distribution. All evaluated models preserve global sparsity structure, but diverge in activation content.

Surprisingly, when assessed at the level of binary structure, diffusion models approximate the global sparsity of the natural co-occurrence matrix with high fidelity. As shown in Fig. 19 (left), the $\ell_0$ norm of $\boldsymbol{Z}^{\mathsf{T}}\boldsymbol{Z}$ – thresholded at varying $\epsilon$ values – tracks closely between the natural and synthesized distributions across all models. This indicates that the gross connectivity of the conceptual graph, i.e., which concepts tend to co-activate at all, is well preserved. Formally, one can deem $\boldsymbol{Z}^{\mathsf{T}}\boldsymbol{Z}$ as the adjacency matrix of a weighted, undirected graph over concepts, where edge weights reflect co-activation strength across the dataset.

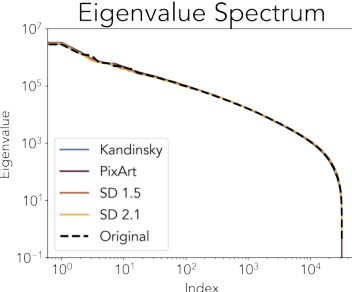

Figure 20: ● **Spectral Structure of Co-occurrence.** Log-log plot of the eigenvalue spectra from co-occurrence matrices $ZZ^T$ across models and the natural distribution. All evaluated models match the heavy-tailed decay of the natural distribution.

However, as illustrated in Fig. 19 (right), the specific content of these co-activations diverges: a substantial portion of entries in the model-generated $\boldsymbol{Z}^{\mathsf{T}}\boldsymbol{Z}$ are not shared with the natural baseline. This suggests that while the capacity for compositionality is retained, the identity of active pairings may shift, potentially reflecting model specific inductive biases or training artifacts. To probe the internal structure of these co-occurrence patterns, we turn to spectral analysis. Fig. 20 shows the eigenvalue spectra of the co-occurrence matrices for each model and the natural distribution. All spectra exhibit a heavy-tailed decay, consistent with power-law behavior, indicating that generative models preserve the overall rank structure and variance allocation across conceptual dimensions.

Further, we examine the alignment of dominant conceptual directions via cosine similarity heatmaps between the top 100 eigenvectors of the synthesized and natural co-occurrence matrices (Fig. 21).

Figure 21: ● **Concept Basis Similarity.** Cosine similarity heatmaps between the top 100 eigenvectors of the natural and synthesized co-occurrence matrices $ZZ^T$. Diagonal structure shows alignment of dominant conceptual directors, with varying degrees of alignment across the four models.

While all models exhibit partial diagonal alignment—implying overlap in principal concept axes—the off-diagonal entries reveal rotations and mismatches in higher modes, reflecting evident deviations in compositional geometry.

Together, these findings reveal that diffusion models approximate the global shape of concept co-activation surprisingly well, yet deviate in subtle and structured ways when examined through the spectral lens. Such higher-order discrepancies may underpin failures in generating coherent, multi-object scenes or relational concepts.

# I    ADDITIONAL RESULTS: CAPTION NOISE IN HIGH DIVERGENCE

We inspected datapoints with the highest datapoint-level energy differences to determine whether the divergence stems from genuine blindspots or low-quality input data (specifically, noisy captions).

## I.1    METHODOLOGY

We isolated the top $k$ datapoints with the highest datapoint-level energy difference $\|\xi(\boldsymbol{x}') - \xi(\boldsymbol{x})\|_2$ (see Section 4.4). We analyzes these using a Vision-Language Model acting as an AutoRater (also referred to as "LLM-as-a-judge"). The VLM was presented with the natural image $\boldsymbol{x}$ and the text prompt $\boldsymbol{t}$, and tasked with classifying the caption quality into three categories:

1. **Noisy.** The caption is irrelevant, factual nonsense, or consists purely of meta-data (e.g., filenames, URLs).
2. **Underspecified.** The caption is technically correct but too vague to identify the specific concepts visible in the image.
3. **Sufficient.** The caption provides enough semantic detail that a generative model should reasonably be expected to reproduce the main concepts visible in the image.

## I.2    RESULTS

We used ChatGPT-4o as the VLM and analyzed $k=199$ images. Out of these, 76 (38.2%) were labeled as noisy, 1 (5.5%) as underspecified, and 112 (56.3%) as sufficient.

## I.3    ANALYSIS

While a portion of the highest energy differences are indeed attributable to poor instruction quality, inherent in web-scraped datasets like LAION-5B, the majority of cases have a sufficient caption. In these instances, the prompt adequately describes the concept present in the natural image, yet the generative model produces a high-energy difference output. This confirms that while dataset noise is a contributing factor, the primary driver of high conceptual divergence remains structural model limitations, supporting the findings in Section 4.4.

## J  ADDITIONAL RESULTS: SAE ERROR CASES

To better understand the specificity and reliability of the SAE used in our experiments to give $\xi(x)$ and $\xi(x')$, we sought to quantify its misfires (false positives, FP) and missing concepts (false negatives, FN).

### J.1  METHODOLOGY

We randomly sampled $n$ AI-generated and $n$ natural images. We constructed the energy profile ($\xi(x)$ or $\xi(x')$) of each image using the evaluated SAE, and filtered for the top-$k$ concepts with the highest activation values. Each concept was mapped to its autointerpretability label.

We then employed a Vision-Language Model (VLM) as an AutoRater (also referred to as "LLM-as-a-judge"). The VLM was presented with the image ($x$ or $x'$) and the list of top-$k$ detected concept descriptions (including their activation strengths), and tasked with two classification objectives:

1. **Misfire (FP) Detection.** Identify concepts in the detected list that are *not* visually present in the image.
2. **Missing Concept (FN) Detection.** Identify critical visual concepts clearly present in the image but *absent* from the detected concept list.

The VLM was instructed to weigh activation strength when evaluating misfires, as concepts with very low activations are more likely to be spurious detections.

### J.2  RESULTS

We conducted the analysis on the top-$k=20$ concepts from $n=100$ natural images and $n=100$ AI-generated images (namely generated using SD 1.5), using ChatGPT-4o as the VLM. The top misfires for natural and AI-generated images are reported in Tables 2 and 3, respectively. Structural concepts with incorrect autointerpretability labels (see Section J.3) are shown in italics.

| #  | Concept | Count |
|----|---------|-------|
| 1  | *(Striped marine creatures)* | *98* |
| 2  | *(Human interaction with surroundings)* | *20* |
| 3  | Natural textures and organic forms | 17 |
| 4  | Green apples and playful animals | 9 |
| 5  | Musical instruments and accessories | 7 |
| 6  | Objects and symbols representing time | 6 |
| 7  | Airships and vintage photography | 5 |
| 8  | Red curtains and theatrical elements | 4 |
| 9  | Airplanes and clothing textures | 4 |
| 10 | Elegant fashion models in glamorous... | 4 |

| #  | Concept | Count |
|----|---------|-------|
| 1  | *(Striped marine creatures)* | *100* |
| 2  | *(Human interaction with surroundings)* | *18* |
| 3  | Natural textures and organic forms | 13 |
| 4  | Elegant fashion models in glamorous... | 9 |
| 5  | Musical instruments and everyday... | 4 |
| 6  | Airplanes and clothing textures | 4 |
| 7  | Bathroom fixtures and sinks | 3 |
| 8  | Leather couch and animals | 3 |
| 9  | Smoke and vapor emissions | 3 |
| 10 | Bookshelves and seating arrangements | 3 |

Table 2: **Top Misfires for Natural Images.** Concepts from the top-$k=20$ concepts of natural images, labeled as misfires in a VLM AutoRater analysis (conducted using ChatGPT-4o). *Concepts in cursive* are structural concepts with inaccurate autointerpretability labels.

Table 3: **Top Misfires for AI Images.** Concepts from the top-$k=20$ concepts of AI-generated images (SD 1.5), labeled as misfires in a VLM AutoRater analysis (conducted using ChatGPT-4o). *Concepts in cursive* are structural concepts with inaccurate autointerpretability labels.

Figures 22 and 23 show the cumulative misfire rate as a function of $k$. Figure 23 excludes structural concepts with incorrect autointerpretability labels, since these are not visibly present in the images and thus cannot be correctly annotated by the VLM; Figure 22 includes all top-$k$ concepts.

Only $8.0\%$ (AI-generated) and $11.1\%$ (natural) of images had critical visual concepts missing from the top-20. The share of concepts that were misfires among the top-20 was $24.04\%$ for natural images and $26.10\%$ for AI-generated images.

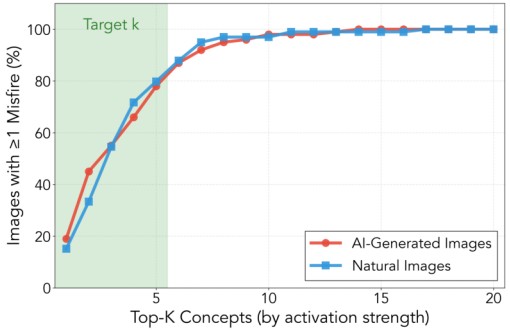

Figure 22: **Cumulative Concept Misfire (All).** Share of the $n{=}100$ images with at least one concept misfire, labeled in a VLM AutoRater analysis (conducted using ChatGPT-4o), as a function of $k$. All concepts are included, including structural ones which are not visible and where the autointerpretability description is inaccurate. The region highlighted as "Target $k$" corresponds to the $k$ hyperparameter of the SAE.

Figure 23: **Cumulative Concept Misfire (Visible Concepts Only).** Share of the $n{=}100$ images with at least one concept misfire, labeled in a VLM AutoRater analysis (conducted using ChatGPT-4o), as a function of $k$. Structural concepts that are not visible and where the autointerpretability description is inaccurate are not included. The region highlighted as "Target $k$" corresponds to the $k$ hyperparameter of the SAE.

### J.3 ANALYSIS

Tables 2 and 3, listing the top misfires in natural and AI-generated images, point to a phenomenon where abstract concepts fire frequently without being tied to a clear semantic feature visible in the image. As shown in Figures 24 and 25, these concepts attend to abstract content near the left or right edges of the image, without any particular semantic attachment. Consequently, the autointerpretability descriptions assigned to these concepts by a VLM (e.g., `Striped marine creatures` and `Human interaction with surroundings`) are not truly descriptive. The VLM AutoRater then marks these concepts as misfires for almost every image because it relies purely on those textual descriptions (Figure 22).

After filtering out such concepts, the misfire profile in Figure 23 shows strong performance under the SAE configuration with $k{=}5$. Here, the SAE activates only five sparse concept codes per image, and these are the positions that should be primarily scrutinized. At $k{=}5$, for both natural and AI-generated images, the majority of datapoints exhibit no misfires. This rate increases approximately linearly until it plateaus around $k{=}20$. Furthermore, only $8.0\%$ of AI-generated images and $11.1\%$ of natural images had critical visual concepts missing from the top-20 concepts.

Specific examples of datapoints with misfires or missing concepts for both natural and AI-generated images are given in Sections J.3.1 and J.3.2, respectively.

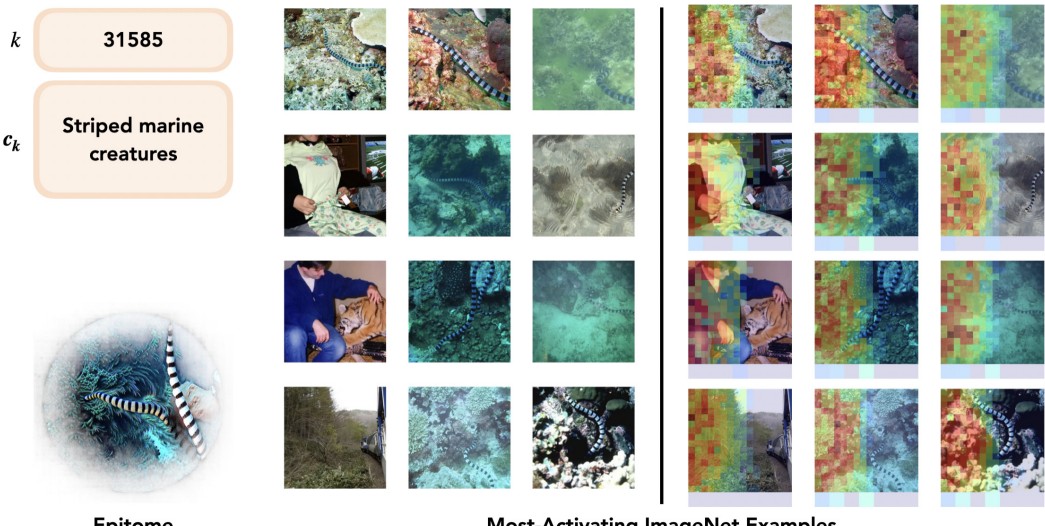

Figure 24: **Concept 31585 Detail.** Shown above are the autointerpretability description, exemplars, and epitome for the concept.

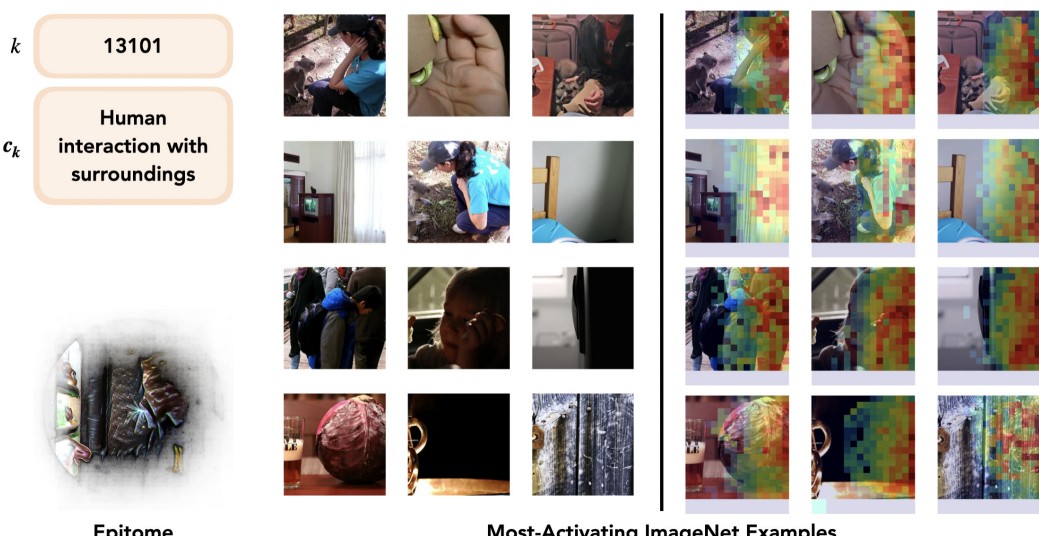

Figure 25: **Concept 13101 Detail.** Shown above are the autointerpretability description, exemplars, and epitome for the concept.

### J.3.1    ERROR CASES: NATURAL IMAGES

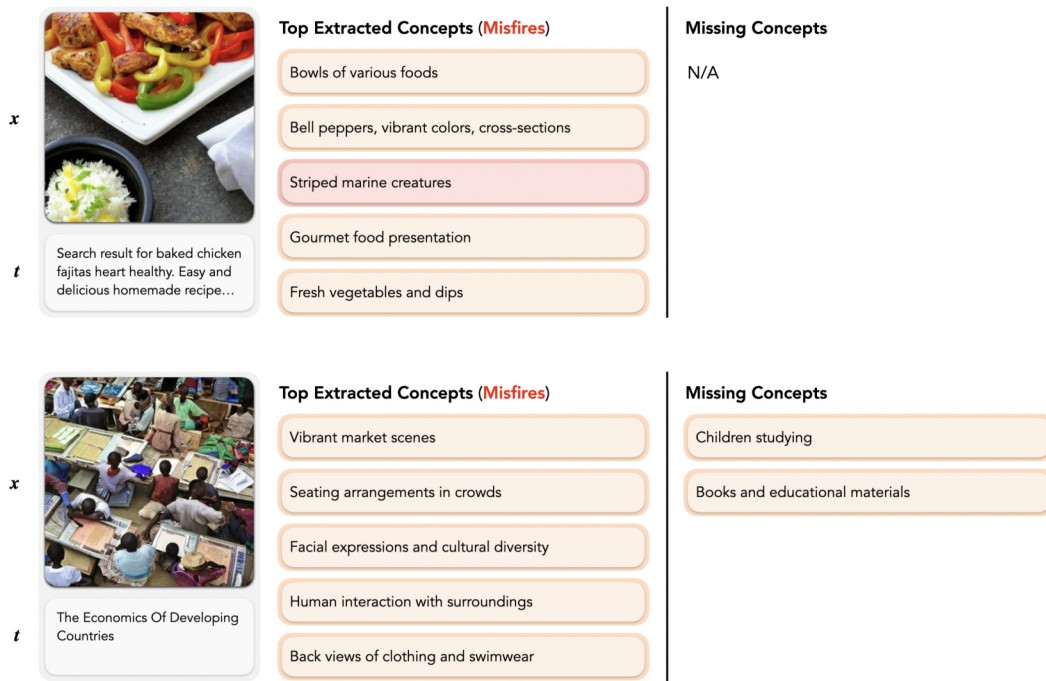

### J.3.2    ERROR CASES: AI-GENERATED IMAGES

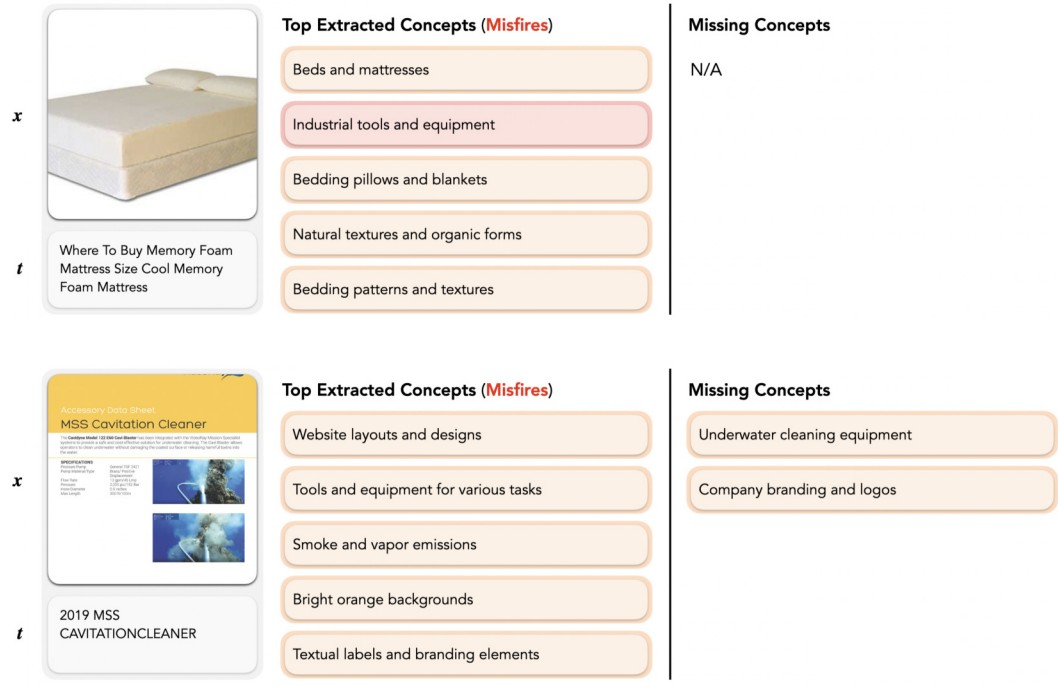

# K CONCENTRATION BOUNDS FOR $\delta$

In our experiments, we estimate $\delta(k)$ using $n = 10{,}000$ paired samples for each concept. While this budget is modest, it raises the natural question of whether it suffices to obtain reliable estimates. To address this, we derive a concentration bound on the empirical estimator $\widehat{\delta}_n(k)$ using McDiarmid's inequality McDiarmid et al. (1989). The resulting bound is tight and demonstrates that even with relatively few samples, we can obtain fast and accurate estimates of concept bias.

**Theorem 4** (Concentration of $\widehat{\delta}_n(k)$)**.** *We assume that the concept score $\xi_k(\boldsymbol{x})$ takes values in $[a, b]$ almost surely for all images $\boldsymbol{x}$ drawn from either $\mathcal{D}_{\mathcal{X}}$ or $\mathcal{D}'_{\mathcal{X}}$. Let $n$ paired samples $(\boldsymbol{x}_i, \boldsymbol{x}'i)_{i=1}^n$ be drawn independently with $\boldsymbol{x}_i \sim \mathcal{D}_{\mathcal{X}}$ and $\boldsymbol{x}'_i \sim \mathcal{D}'_{\mathcal{X}}$, and define the empirical estimator*

$$\widehat{\delta}_n(k) := \sigma\left(\frac{1}{n}\sum_{i=1}^n \xi_k(\boldsymbol{x}_i) - \frac{1}{n}\sum_{i=1}^n \xi_k(\boldsymbol{x}'_i)\right).$$

*Let $M := b - a$ and $L := M/4$. Then for every $\varepsilon > 0$, the deviation satisfies*

$$\mathbb{P}\left(\left|\widehat{\delta}_n(k) - \delta(k)\right| > \varepsilon\right) \leq 2\exp\left(-\frac{2n\varepsilon^2}{L^2}\right).$$

*Proof.* The function $x \mapsto \sigma(x)$ is $1/4$-Lipschitz, since $|\sigma'(x)| \leq 1/4$ for all $x$.

Viewing $\widehat{\delta}_n(k)$ as a function of the $2n$ independent variables $(\boldsymbol{x}_1, \ldots, \boldsymbol{x}_n, \boldsymbol{x}'_1, \ldots, \boldsymbol{x}'_n)$, changing a single argument alters the inner difference of means by at most $M/n$, and the outer sigmoid scales this by at most $1/4$. Hence, the bounded difference constant for each coordinate is $(M/n)(1/4) = L/n$.

By McDiarmid's inequality (McDiarmid et al., 1989),

$$\mathbb{P}\left(\left|\widehat{\delta}_n(k) - \delta(k)\right| > \varepsilon\right) \leq 2\exp\left(-\frac{2\varepsilon^2}{\sum_{j=1}^{2n}(L/n)^2}\right) = 2\exp\left(-\frac{2n\varepsilon^2}{L^2}\right),$$

which proves the claim. $\square$

Practically, most concept scores $\delta(k)$ are sparse, with the majority concentrated near zero and only a few reaching values up to 10. The concentration bound shows that even for the largest observed biases, a sample size of $n = 10{,}000$ yields estimates of $\widehat{\delta}_n(k)$ that deviate from the true value by no more than a small $\varepsilon$ with high probability. This justifies our sampling strategy and confirms that accurate bias measurements are attainable with limited data.

## L    MONOTONICITY AND CALIBRATION-FREE INTERPRETATION OF $\delta$

Our goal when analyzing blind spots is to rank concepts by the severity of their generative bias. In practice, we use the score $\delta(k)$ for this purpose. However, one may wonder whether such a score introduces distortions relative to more direct quantities such as the energy gap or the odds ratio. The following result establishes that $\delta(k)$ is a strictly increasing reparameterization of both, and therefore inherits their ordering. This guarantees that no calibration is needed when using $\delta(k)$ to rank concepts.

**Theorem 5** (Monotonicity and Calibration of $\delta_{\boldsymbol{g_\theta} \leftrightarrow \boldsymbol{G}}$). *For every concept index $k$ define the energy gap*

$$\Delta_k = \mathbb{E}_{\boldsymbol{x'} \sim \mathcal{D}'_\mathcal{X}}\big[\xi_k(\boldsymbol{x'})\big] - \mathbb{E}_{\boldsymbol{x} \sim \mathcal{D}_\mathcal{X}}\big[\xi_k(\boldsymbol{x})\big],$$

*the associated odds ratio $\rho_k = \exp(\Delta_k)$, and the energy–difference score*

$$\delta(k) \;=\; \frac{1}{1 + \exp(-\Delta_k)} \;=\; \frac{\rho_k}{1 + \rho_k}.$$

*Then $\delta(k)$ is a strictly increasing bijection of both $\Delta_k$ and $\rho_k$, so ranking concepts by any one of $\delta(k)$, $\Delta_k$, or $\rho_k$ produces exactly the same ordering.*

*Proof.* The logistic sigmoid satisfies $\sigma'(x) = \sigma(x)\big(1 - \sigma(x)\big) > 0$, $\forall x \in \mathbb{R}$; hence $\sigma$ and therefore $\delta(k) = \sigma(\Delta_k)$ grow strictly with $\Delta_k$. Because the exponential map is also strictly increasing and bijective $\mathbb{R} \to (0, \infty)$, setting $\rho_k = \exp(\Delta_k)$ preserves order and gives $\Delta_k = \log \rho_k$. Substituting this identity into $\sigma$ yields $\delta(k) = \sigma(\log \rho_k) = \rho_k/(1 + \rho_k)$, which is the composition of two strictly increasing bijections and is therefore itself strictly increasing and bijective in $\rho_k$. Since strict monotonic functions never reverse inequalities, the three quantities share the same total order over concepts. $\qquad\square$

Thus, ranking concepts by $\delta(\cdot)$ is strictly equivalent to ranking them by energy gap or by conceptual generation odds $\rho_k$. No calibration is necessary, and all three quantities preserve the same total ordering over concepts.

## M  STABILITY OF FID UNDER SAE EMBEDDINGS

In this section we establish a quantitative relationship between the Fréchet Inception Distance (FID) computed in the original activation space (of dimension $d$) and the FID after applying a (potentially overcomplete) SAE dictionary $\boldsymbol{D} \in \mathbb{R}^{k \times d}$ with $k \gg d$. Throughout we assume that $\boldsymbol{D}$ has orthonormal columns but is not necessarily square, i.e.

$$\boldsymbol{D}^\top \boldsymbol{D} = \boldsymbol{I}_d, \qquad \text{while} \qquad \boldsymbol{D}\boldsymbol{D}^\top \neq \boldsymbol{I}_k.$$

We start by recalling a simple fact: if $\boldsymbol{D}$ is not overcomplete, orthogonal and $k = d$, the we have an isometry between $\boldsymbol{A}$ and $\boldsymbol{Z}$, implying that the FID is perfectly preserved. However, this case is not realistic, we will then turn the the overcomplete case, and show we can bound FID by the extreme singular value of $\boldsymbol{D}$. We will work with the Wasserstein-2 metric $\mathcal{W}_2$, noting that FID is just $\mathcal{W}_2^2$ specialised to Gaussians.

For a probability measure $\boldsymbol{\mu}$ on $\mathbb{R}^d$ we write $\boldsymbol{D}_{\#}\boldsymbol{\mu}$ for its push-forward under $\boldsymbol{D}$, i.e. $\boldsymbol{D}_{\#}\boldsymbol{\mu}(\boldsymbol{z}) = \boldsymbol{\mu}(\boldsymbol{D}^{-1}\boldsymbol{z})$. Denote by $\sigma_{\min}$ and $\sigma_{\max}$ the minimal and maximal singular values of $\boldsymbol{D}$, equivalently the square-roots of the extremal eigenvalues of $\boldsymbol{D}\boldsymbol{D}^\top$:

$$\sigma_{\min}^2 \boldsymbol{I}_k \preceq \boldsymbol{D}\boldsymbol{D}^\top \preceq \sigma_{\max}^2 \boldsymbol{I}_k.$$

Empirically one usually finds $\sigma_{\min}, \sigma_{\max} \approx 1$, but the proof does not rely on that. We will start by a simple lemma in the case where $\boldsymbol{D}$ is not overcomplete.

**Lemma 1** (Isometry under exact orthogonality)**.** *Suppose $k = d$ and $\boldsymbol{D}^\top \boldsymbol{D} = \boldsymbol{D}\boldsymbol{D}^\top = \boldsymbol{I}_d$. Then $\boldsymbol{D}$ is an isometry: $\|\boldsymbol{D}\boldsymbol{v}\|_2 = \|\boldsymbol{v}\|_2$ for all $\boldsymbol{v} \in \mathbb{R}^d$. Consequently, for any probability measures $\boldsymbol{\mu}, \boldsymbol{\nu}$ on $\mathbb{R}^d$ with finite second moment,*

$$\mathcal{W}_2\big(\boldsymbol{D}_{\#}\boldsymbol{\mu}, \boldsymbol{D}_{\#}\boldsymbol{\nu}\big) = \mathcal{W}_2(\boldsymbol{\mu}, \boldsymbol{\nu}).$$

*Proof.* Orthogonality of $\boldsymbol{D}$ implies preservation of the Euclidean norm, and push-forward commutes with the map inside the $\mathcal{W}_2$ infimum; the integrand is unchanged, so the infimum value is identical. $\square$

This case, however, is quite unrealistic as SAE usually rely on the overcompletness to extract meaningful and interpretable concepts. In the overcomplete case, $\boldsymbol{D}$ is no longer orthonormal, but we can still have column-orthonormal dictionary. We will use that to show that we can bound using the extremal singular value of $\boldsymbol{D}^\top \boldsymbol{D}$.

**Theorem 6** (FID under column orthogonal embeddings)**.** *Let $\boldsymbol{D} \in \mathbb{R}^{k \times d}$ satisfy $\boldsymbol{D}^\top \boldsymbol{D} = \boldsymbol{I}_d$ and denote by $0 < \sigma_{\min} \leq \sigma_{\max}$ the extreme singular values of $\boldsymbol{D}\boldsymbol{D}^\top$. Given two data matrices $\boldsymbol{A}, \boldsymbol{A}' \in \mathbb{R}^{n \times d}$ (rows are sample vectors), set $\boldsymbol{Z} = \boldsymbol{A}\boldsymbol{D}^\top \in \mathbb{R}^{n \times k}$ and $\boldsymbol{Z}' = \boldsymbol{A}'\boldsymbol{D}^\top \in \mathbb{R}^{n \times k}$. Then*

$$\sigma_{\min}^2 \, \mathrm{FID}(\boldsymbol{A}, \boldsymbol{A}') \leq \mathrm{FID}(\boldsymbol{Z}, \boldsymbol{Z}') \leq \sigma_{\max}^2 \, \mathrm{FID}(\boldsymbol{A}, \boldsymbol{A}').$$

*Proof.* Write $\boldsymbol{\mu}$ for the empirical measure of $\boldsymbol{A}$ and $\boldsymbol{\nu}$ for that of $\boldsymbol{A}'$, i.e.

$$\boldsymbol{\mu} = \frac{1}{n} \sum_{i=1}^{n} \boldsymbol{\delta}_{\boldsymbol{A}_{i,:}}, \qquad \boldsymbol{\nu} = \frac{1}{n} \sum_{i=1}^{n} \boldsymbol{\delta}_{\boldsymbol{A}'_{i,:}}.$$

For any coupling $\boldsymbol{\pi} \in \Pi(\boldsymbol{\mu}, \boldsymbol{\nu})$ (i.e. a probability measure on $\mathbb{R}^d \times \mathbb{R}^d$ with marginals $\boldsymbol{\mu}, \boldsymbol{\nu}$) we have, by the extremal singular value bound,

$$\sigma_{\min}^2 \|\boldsymbol{x} - \boldsymbol{y}\|_2^2 \leq \|\boldsymbol{D}(\boldsymbol{x} - \boldsymbol{y})\|_2^2 \leq \sigma_{\max}^2 \|\boldsymbol{x} - \boldsymbol{y}\|_2^2, \qquad \forall (\boldsymbol{x}, \boldsymbol{y}) \in \mathbb{R}^d \times \mathbb{R}^d.$$

Integrating with respect to an arbitrary coupling $\boldsymbol{\pi} \in \Pi(\boldsymbol{\mu}, \boldsymbol{\nu})$ yields

$$\sigma_{\min}^2 \int \|\boldsymbol{x} - \boldsymbol{y}\|_2^2 \, d\boldsymbol{\pi} \leq \int \|\boldsymbol{D}(\boldsymbol{x} - \boldsymbol{y})\|_2^2 \, d\boldsymbol{\pi} \leq \sigma_{\max}^2 \int \|\boldsymbol{x} - \boldsymbol{y}\|_2^2 \, d\boldsymbol{\pi}.$$

The middle integral is exactly the transport cost of the pushed–forward coupling $(D \times D)_{\#}\boldsymbol{\pi}$ between $\boldsymbol{\mu}_D := D_{\#}\boldsymbol{\mu}$ and $\boldsymbol{\nu}_D := D_{\#}\boldsymbol{\nu}$. Because the inequalities hold for *every* $\boldsymbol{\pi}$, they hold in particular

for the optimal couplings attaining $\mathcal{W}_2(\boldsymbol{\mu}, \boldsymbol{\nu})$ and $\mathcal{W}_2(\boldsymbol{\mu}_D, \boldsymbol{\nu}_D)$, though these two optima need not coincide. Taking the infimum over $\boldsymbol{\pi}$ term-wise makes this explicit:

$$\sigma_{\min}^2 \inf_{\boldsymbol{\pi} \in \Pi(\boldsymbol{\mu}, \boldsymbol{\nu})} \int \|\boldsymbol{x} - \boldsymbol{y}\|_2^2 \, d\boldsymbol{\pi} \ \leq \ \inf_{\boldsymbol{\pi} \in \Pi(\boldsymbol{\mu}, \boldsymbol{\nu})} \int \|D(\boldsymbol{x} - \boldsymbol{y})\|_2^2 \, d\boldsymbol{\pi} \ \leq \ \sigma_{\max}^2 \inf_{\boldsymbol{\pi} \in \Pi(\boldsymbol{\mu}, \boldsymbol{\nu})} \int \|\boldsymbol{x} - \boldsymbol{y}\|_2^2 \, d\boldsymbol{\pi}.$$

Hence

$$\sigma_{\min}^2 \, \mathcal{W}_2^2(\boldsymbol{\mu}, \boldsymbol{\nu}) \ \leq \ \mathcal{W}_2^2(\boldsymbol{\mu}_D, \boldsymbol{\nu}_D) \ \leq \ \sigma_{\max}^2 \, \mathcal{W}_2^2(\boldsymbol{\mu}, \boldsymbol{\nu}).$$

Recognising $\mathrm{FID}(\cdot, \cdot) = \mathcal{W}_2^2(\cdot, \cdot)$ for the Gaussian surrogate and plugging in $(\boldsymbol{A}, \boldsymbol{A}')$ (resp. $(\boldsymbol{Z}, \boldsymbol{Z}')$) finishes the proof. □

Essentially, theorem 6 tells us that applying a column-orthogonal overcomplete SAE dictionary cannot distort Fréchet Inception Distance by more than the square of its extremal singular values. When $\boldsymbol{D}$ is nearly orthogonal – empirically we usually found that $\sigma_{\min}, \sigma_{\max} \approx 1$ – the result implies that FID measured in the SAE feature space is essentially close to the canonical FID. □

# N ADDITIONAL EXAMPLES OF SYNTHESIZED IMAGES

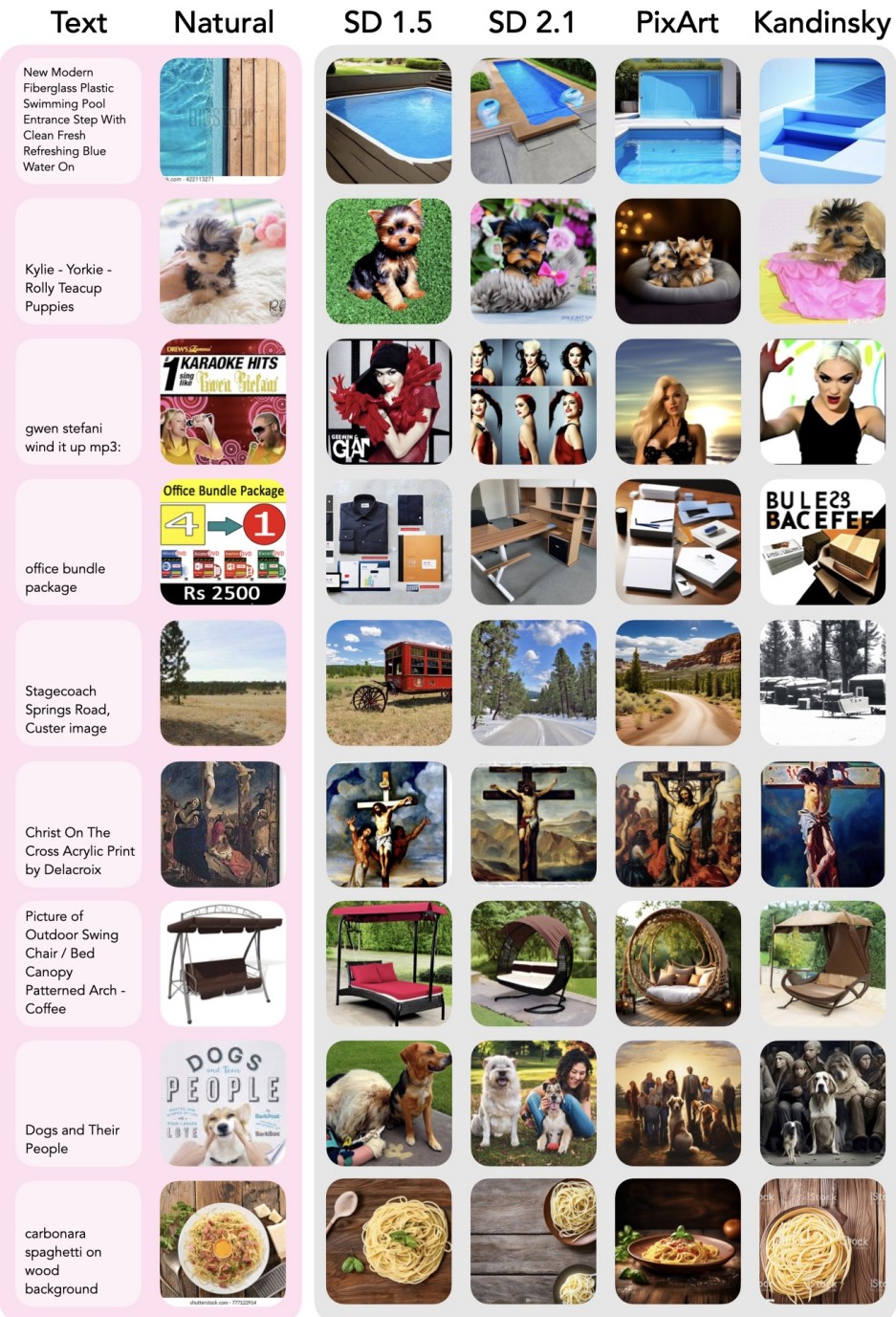

Figure 26: Additional image-caption pair examples from LAION-5B with matching images generated with the same prompt by SD 1.5, SD 2.1, PixArt, and Kandinsky.

