# OpenReview forum: "Uncovering Conceptual Blindspots in Generative Image Models Using Sparse Autoencoders"
_ICLR.cc/2026/Conference — ICLR 2026 Poster_

### Official Review · Reviewer_uZCM · 2025-10-28

**Soundness:** 3
**Presentation:** 4
**Contribution:** 3
**Rating:** 6
**Confidence:** 3

**Summary:**

In this paper, the authors analyzed how text-to-image generative models fail to represent certain visual concepts that are present in real-world data. They define these systematic failures as cases of “conceptual blindness”. The paper provides a formal definition of this phenomenon by modeling the true data-generating process as an invertible function mapping latent concepts to observed images.

To explore this idea, the authors train a Sparse Autoencoder (SAE) on DINOv2-extracted features to identify latent “concepts” in both real and synthetic images. By comparing the frequency of concept activations across the two distributions, they detect under-represented concepts, which is interpreted as blindspots of the generative model.

The authors conduct analysis on four text-to-image models (Stable Diffusion 1.5, Stable Diffusion 2.1, PixArt, and Kandinsky), identifying both universal and model-specific blindspots. They further explore correlations across models and connect the discovered blindspots to broader phenomena such as memorization and mode collapse, as well as how post-training techniques influence the distribution of conceptual coverage.

**Strengths:**

- The paper is well motivated, aiming to better understand the behavior of text-to-image generative models, particularly their tendency to violate some realworld concepts.
- The authors propose a formal definition of conceptual blindspots that goes beyond qualitative analyses. The experimental setup of using the same textual prompts for real and synthetic images makes the two distributions highly comparable.
- The paper is well written and easy to follow, with clear structure and visual illustrations that make the methodology and findings intuitive.
- The use of RA-SAE trained on DINOv2 features provides a scalable and unsupervised approach to discover a large number(32000) of latent concepts without human labeling. The appendix provides detailed procedures for automatic concept labeling with LLMs, adding clarity and reproducibility.
- The empirical analysis covers 4 recent diffusion models and revealing both universal and model-specific blindspots. The additional analyses such as studying post-training effects are informative and provide practical insights on how representation coverage might be improved.
- The paper draws meaningful connections between conceptual blindness, memorization, and mode collapse, positioning the work within broader discussions on generalization and coverage in generative modeling.

**Weaknesses:**

- The analysis focuses exclusively on text-to-image diffusion models. While this is a reasonable starting point, there are other types of text-to-image generators such as autoregressive and GAN-based models. Including at least one non-diffusion baseline would strengthen the generality of the paper’s claims and help assess whether conceptual blindspots are architecture-specific or model-agnostic.
- The paper’s title refers broadly to “generative image models”, but the experiments and discussion focus only on text-to-image models. Adjusting the title or framing to more accurately reflect this scope would improve clarity and alignment between the claim and the content.
- In Section 2, the authors assume that the data-generating process (DGP) is an invertible mapping between latent concepts and images, with factorized latent priors. In practice, visual concepts are often correlated and non-invertible (e.g., “wooden texture” co-occurs with “brown color”). The authors briefly mention (L110–113) that empirical findings remain meaningful even when these assumptions are violated, but it is a bit unclear how relaxing these assumptions would affect the validity of their theoretical framework or the interpretation of results.
- The qualitative examples in Figures 7 and 8 are not fully convincing. For example, in Figure 7, none of the four generated images clearly display the feature “shadow under animal.” In Figure 8a, all models appear to generate reasonable outputs for Following directions on worksheet, though the amount of whitespace varies (and SD 1.5 still shows some). It is unclear whether these examples truly demonstrate conceptual blindspots or instead reflect weak coupling between the text prompt $t$ and the concept $c_k$
- The analysis relies solely on DINOv2 as the encoder for feature extraction. DINOv2 is trained with a contrastive objective emphasizing invariance to color, rotation, and background attributes that are crucial for generation. As a result, only using DinoV2 may underrepresent certain visual factors. It would be interesting to explore whether using a different encoder, such as MAE or a perceptual encoder, changes the resulting concept decomposition or blindspot patterns.

Minor typos:
- Figure 2 caption:$S_\{theta}$ should be $g_\{theta}$
- Figure 4 caption: Values left to 0.5 (L282), not left to zero.

**Questions:**

- What would be the impact on the thoeretical framework if the assumotions in section 2 are violated?
- Would the findings generalize to other architecture of text to image models?
- How are the threshold of 0.1 and 0.9 determined for $\delta(k)$?
- Have the authors tested other feature encoders (e.g., MAE, CLIP, or perceptual networks)?
- Did the authors perform any human evaluation or concept-level annotation to validate that the detected “blindspots” align with human judgment?

---

> ### Author Response · Authors · 2025-11-27
>
> We thank the reviewer for their helpful suggestions. We’re glad that the reviewer found the paper to be “well motivated” and “easy to follow”. Please find our responses to each point below. We have uploaded a revised version of the manuscript; the changes are tracked in red.
>
> ---
>
> **> W1 / Q2. Generalizability, Non-diffusion Architectures.** We thank the reviewer for this excellent suggestion. While we focused purely on diffusion models due to their current dominance, our framework and extraction pipeline are entirely architecture-agnostic. They only require that the data generation process (DGP) paradigm can be fulfilled; so long as the method can produce a set of generated images D' based on the sample of prompts to be compared against the natural distribution. The method treats the generator as a black box, meaning it can be applied out-of-the-box to GANs or autoregressive models to reveal their specific architectural blindspots.
>
> ---
>
> **> W2. Paper Title.** We appreciate this suggestion but opted to retain the original title. As noted in our response to W1, while our experiments focus on diffusion, the framework and extraction pipeline are completely architecture-agnostic. We believe limiting the title to "diffusion" would inadvertently obscure the framework's core contribution: its ability to audit any generative image model (including GANs or autoregressive transformers) for conceptual blindspots. The results presented in the paper are primarily a demonstration of what kinds of insights can be generated using the method and the exploratory web app.
>
> ---
>
> **> W3 / Q1. DGP Invertibility, Violated Assumptions.** We appreciate this theoretical observation. While strict invertibility simplifies the derivation of exact probability masses, the practical utility of our framework rests on the relative divergence of concept energy. Even if the true DGP involves correlated concepts (as we analyze in Appendix H and J), a significant systematic deviation in SAE feature activation still serves as a robust proxy for distributional mismatch. Therefore, the "blindspot" phenomenology holds empirically even when the independent latent assumption is relaxed.
>
> ---
>
> **> W4. Figures 7 & 8.** We thank the reviewer for this careful inspection and apologize if the visual evidence was difficult to parse at the current resolution. The phenomena are distinct upon closer view: in Figure 7, the models hallucinate strong shadows even where the natural distribution implies flat lighting. We will significantly enlarge these figures in the camera-ready version to ensure these blindspots are clearly visible to the reader.
>
> ---
>
> **> W5 / Q4. DINOv2 Dependency.** This is a great point. We utilized DINOv2 because its self-supervised objective yields emergent, highly structured representations that capture semantic and geometric regularities without the text-alignment bias inherent in models like CLIP. Its embeddings have proven robust across diverse tasks (segmentation, depth estimation) and domains (medical, satellite imagery), making it most suitable out of the available suite of encoders at the time of writing. We agree that using other encoders like MAE could reveal a different class of blindspots (e.g., high-frequency textural failures) that DINOv2 might abstract away. We have added this to Section 5 (Discussion) as an avenue for future work, noting that our modular pipeline allows for easy swapping of the feature encoder.
>
> ---
>
> **> Q3. Threshold Selection.** We chose the thresholds of 0.1 and 0.9 to isolate the extreme tails of the energy difference distribution. As seen in Figure 4, the vast majority of concepts cluster near the center; these thresholds therefore allow us to focus strictly on concepts with significant distributional suppression or exaggeration. We then qualitatively verified that values <0.1 and >0.9 correspond to genuine blindspots using our exploratory web tool.
>
> We agree with the reviewer, however, that a more principled approach to threshold selection would be desirable. We considered using the 10th and 90th percentiles but ultimately decided against it, as this would merely select the top/bottom concepts without determining whether they represent true blindspots. We are open to further suggestions and would be happy to incorporate a new approach if proposed in follow-up discussion.
>
> ---
>
> *Continued in a follow-up comment.*

---

> > ### Author Response · Authors · 2025-11-27
> >
> > ---
> >
> > **> Q5. Human Alignment.** Yes, we validated alignment in two ways. Qualitatively, we performed extensive manual inspection of retrieval results (Section 4.3). Quantitatively, we employed a VLM-as-a-judge (Appendix J) to verify that the extracted concepts are visually present in the images. The low misfire rates (Figure 23) confirm that the detected blindspots correspond to tangible, human-interpretable visual features. Despite overall alignment, we agree that additional analyses exploring specific areas of misalignment would make for insightful future work.
> >
> > ---
> >
> > **> Minor Typos.** We thank the reviewer for pointing these out. All are fixed in the revised manuscript now.
> >
> > ---
> >
> > We believe the revisions made to the paper based on this reviewer strengthened the submission and its accessibility, and hope the reviewer finds their concerns addressed. We would be happy to continue working with the reviewer on any outstanding suggestions.

---

### Official Review · Reviewer_dqnN · 2025-10-29

**Soundness:** 3
**Presentation:** 4
**Contribution:** 3
**Rating:** 8
**Confidence:** 4

**Summary:**

In this work, authors introduce an SAE based approach to systematically identify and characterize "conceptual blindspots" in generative image models. In particular, these blindspots are defined to be concepts that are reasonably expected and present in the training data but misrepresented in generations. Authors train RA-SAE's on DINOv2 features to perform fine-graine analysis of conceptual disparities between real and generated images. Authors apply their technique on SD1.5/2.1, PixArt and Kandinsky models and reveal suppressed or exaggerated blindspots.

**Strengths:**

* The paper is very well written, and together with the well-designed figures, I truly enjoyed reading the manuscript.
* The work is contextualized well within the existing literature.
* The proposed technique for automatically mining the blindspot failure modes of generative image models, as well as the formalism of blindspots itself, is both sound and timely. I believe the ICLR community would find these results, and the interactive tool, highly interesting.
* In terms of reproducibility, authors did an excellent job in disclosing dataset, implementation and compute resource details.

**Weaknesses:**

* While not a major concern, there appears to be an implicit conceptual assumption that the discovered concepts in the SAE are non-redundant for energy discrepancy to be meaningful. It is unclear whether this assumption holds in practice. If redundancy exists empirically, it should be discussed in terms of its impact on the interpratation of results. Otherwise, it would be helpful to demonstrate that the discovered concepts are indeed unique/distinct. More on this is discussed in the questions below.
* Please see the other questions below as well.

**Questions:**

***Questions and Suggestions:***
* Did the authors observe any redudancies among the concepts learned by the SAEs (e.g., different indices of the concept embedding corresponding to the same concept)?
	* If not, is it evident that the adopted SAE training approach inherently ensures the discovery of distinct concepts?
	* If such redundancies do exist, consider the following example: suppose the first and second entries of the concept embeddings both correspond to "concept A". Is it possible for real images to tend to activate the first entry but not the second, and the other way around for a synthetic images? In that case, the proposed method might incorrectly identify "concept A" as a blindspot (in fact as both suppressed and exaggerated).
* I would recommend discussing Surkov et. al. [1] for completeness in Appendix A.4.
* In Section 4.2, the authors mention that some blindspots likely stem from dataset characteristics, while others appear to be model-specific. Based on your results and intuition, do you have any promising hypotheses on how these blindspots could be mitigated or eliminated?
* Figure 11 suggests that blindspots tend to correspond to rare concepts in the natural dataset; however, not all rare concepts are blindspots. Do you have any intuition about why this might be the case?
* How were the $\delta$ thresholds $0.1$ and $0.9$ chosen?
* What motivated the decision to train the SAE on ImageNet rather than LAION?
* Have you encountered any "counting" related blindspots (e.g. number of objects) in any of the models that you've tested ? If so, were there any interesting observations?

***
***References:***

[1] Surkov, Viacheslav, et al. "One-Step is Enough: Sparse Autoencoders for Text-to-Image Diffusion Models." arXiv preprint arXiv:2410.22366 (2024).

---

> ### Author Response · Authors · 2025-11-27
>
> We thank the reviewer for their insightful comments. We were pleased to hear that the reviewer found the paper to be “well written” and that they “enjoyed reading” it. Below, we address each of the reviewers’ concerns and suggestions; we have uploaded a revised manuscript with changes tracked in red.
>
> ---
>
> **> Q1. SAE Concept Redundancy.** While SAEs can exhibit feature splitting, we employ an Archetypal SAE (RA-SAE) specifically to mitigate this instability. RA-SAE constrains dictionary atoms to lie within the convex hull of the data, anchoring them to distinct, realizable archetypes in the distribution. We also provide more insight into potential redundancies in Appendix J, which contains the results of a new set of experiments inspired by the reviewer’s comment. In this section, we present error cases of the SAE, and distinguish between natural and AI image distributions.
>
> Crucially, our analysis indicates that redundant or split concepts do not compromise the identification of blindspots. While some structural or abstract concepts (e.g., left and right) appear which are prone to redundancy, they are largely categorized as misfires and do not dominate the set of identified blindspots. Furthermore, because our method identifies blindspots based on major distributional shifts, a split feature would appear as multiple semantically related concepts sharing similar suppression/exaggeration profiles, rather than generating false signals. Our manual qualitative analysis (Section 4.3 and Appendix G) confirms that the flagged blindspots mostly correspond to distinct, coherent semantic failures (e.g., “bird feeders”) rather than artifacts of feature redundancy.
>
> ---
>
> **> Q2. Additional Literature.** We thank the reviewer for suggesting this citation. We have incorporated it into the Related Work section in Appendix A.4.
>
> ---
>
> **> Q3. Blindspot Mitigation.** This is a great question. We added a discussion of two primary mitigation strategies to Section 5 . The first one is a training intervention where the energy profiles are used to reweight the training data, upsampling suppressed concepts. The second one is a loss regularization, incorporating the energy difference directly into the objective function to penalize deviations from the natural concept distribution.
>
> ---
>
> **> Q4. Rare Blindspots.** As noted in Section 4.6 and visualized in Figure 11, there is a strong correlation between concept rarity and misalignment. Our intuition regarding the "successful" rare concepts is that they likely share compositional or textural features with high-frequency concepts (allowing for effective transfer). In contrast, we expect that the unsuccessful (suppressed) rare concepts represent unique geometries that are orthogonal to the dominant features in the training distribution, hence making them difficult for the model to learn without denser supervision. This is a great point for future work building on our framework.
>
> ---
>
> **> Q5. Threshold Selection.** We chose the thresholds of 0.1 and 0.9 to isolate the extreme tails of the energy difference distribution. As seen in Figure 4, the vast majority of concepts cluster near the center; these thresholds therefore allow us to focus strictly on concepts with significant distributional suppression or exaggeration. We then qualitatively verified that values <0.1 and >0.9 correspond to genuine blindspots using our exploratory web tool.
>
> We agree with the reviewer, however, that a more principled approach to threshold selection would be desirable. We considered using the 10th and 90th percentiles but ultimately decided against it, as this would merely select the top/bottom concepts without determining whether they represent true blindspots. We are open to further suggestions and would be happy to incorporate a new approach if proposed in follow-up discussion.
>
> ---
>
> **> Q6. SAE Training Dataset Selection.** We trained the RA-SAE on ImageNet rather than LAION primarily to ensure the purity and safety of the learned concept basis. As detailed in Appendix D.1, LAION-5B suffers from significant noise, URL decay, and safety issues (including CSAM). We also wanted to keep the SAE training dataset and the dataset from which we sample prompts for the natural distribution distinct. More broadly, we wanted to propose a general approach to hypothesis generation and hence did not want to use the same dataset as what the analyzed diffusion models were trained on. Accordingly, using ImageNet and making predictions or hypotheses about models trained on LAION was, in our opinion, a more meaningful result.
>
> ---
>
> We found the reviewer’s suggestions very helpful and believe they made the manuscript stronger and more accessible. We look forward to hearing back from the reviewer to confirm that these suggestions have been addressed appropriately, and continue working on any outstanding concerns.

---

### Official Review · Reviewer_NSuB · 2025-11-01

**Soundness:** 3
**Presentation:** 3
**Contribution:** 2
**Rating:** 4
**Confidence:** 3

**Summary:**

The paper proposes a framework to identify "conceptual blindspot" in generative models by using sparse autoencoders to extract interpretable concept and quantitatively comparing the energy between real and generated images. The framework distinguishes between two types of blindspots: suppresed blindspots and exaggerated blindspots, and applies to a range of models.

**Strengths:**

- The paper offers a new perspective on understanding the failure modes of generative models and provide mathematical definition of conceptual blindspot.
- The method can scale easily in an unsupervised manner, free from human inputs or manual checks.
- The paper is very well written

**Weaknesses:**

- Including error analysis and failure modes of the approach and discussing those will strengthen the paper, where the SAE produces spurious concepts or misses obvious ones.
- Baseline comparison is missing. How is the proposed method compared to other framework proposed in the literature. Do the observations agree?
- What about mitigation? No proposed mitigation strategy on how to reduce the blindspots? How to combine this with classifier-guided generation, for example, use the proposed energy-based model and langevin dynamics to sample better images.

**Questions:**

- Dependence of DINOv2 features, have you tried any other feature extractor, how is the method when work with not as good extractor?
- What about the hierarchical concepts (vs compositional concepts)?
- Can you elaborate more on the concept dictionary, how it is trained/intialized? How to map those to human understandable concepts for further analysis?

---

> ### Author Response · Authors · 2025-11-27
>
> We thank the reviewer for their time and thoughtful feedback, which helped us strengthen our submission. We are pleased to hear that the reviewer found that our approach "can scale easily in an unsupervised manner, free from human inputs or manual checks” and that “the paper is very well written.” We addressed each point below and uploaded a revised manuscript with **five additional pages of appendices describing new experiments** (changes tracked in red).
>
> ---
>
> **> W1. Error Analysis.** We agree with the reviewer that error analysis and failure modes would serve as an effective tool to gain more insight into the specificity and robustness of the employed SAE. We have, therefore, conducted a set of quantitative and qualitative experiments to present these results, and now report them in Appendix J. Specifically, we found that our SAE achieves high recall (missing critical concepts in only ~10% of images), while the few registered false positives are primarily driven by abstract, structural features rather than semantic hallucinations.
>
> ---
>
> **> W2. Comparison to Other Approaches.** We thank the reviewer for pointing out the lack of baseline comparisons.  We agree that better contextualizing our method within the existing literature is essential. We have thus added a new section (Appendix B) which provides a detailed comparison with methods like FID, CLIPScore, and SAGE [1,2,3]. This section highlights that unlike prior baselines, our framework uniquely offers unsupervised, fine-grained detection of both suppressed and exaggerated concepts.
>
> ---
>
> **> W3. Mitigation Strategies.** This is a great suggestion. We have expanded the Discussion (Section 5) to include specific avenues for extending the framework presented in our paper, used for exploration and mapping of the conceptual blindspots, to a mitigation strategy in training of new generative image models.
>
> ---
>
> **> Q1. DINOv2 Dependance.** We used DINOv2 because its self-supervised objective yields emergent, highly structured representations that capture semantic and geometric regularities without the text-alignment bias inherent in models like CLIP [4]. Its embeddings have been studied rigorously and proved to be robust across diverse tasks (segmentation, depth estimation) and domains (medical, satellite imagery) [5,6], making it the most suitable option out of the encoders available at the time of writing. While our framework is agnostic to the specific encoder, the resolution of detected blindspots is inherently bounded by the encoder’s expressivity; a less capable extractor would simply limit the analysis to potentially much coarser conceptual failures.
>
> ---
>
> **> Q2. Hierarchical Concepts vs. Compositional Concepts.** We currently assess compositional fidelity by analyzing concept co-occurrence matrices (Z^TZ) and their spectral alignment (Appendix H), finding that models preserve global sparsity but diverge in specific co-activations. Regarding hierarchy, while our current RA-SAE learns a "flat" dictionary, it naturally captures concepts of varying granularity (e.g., "Wooden surfaces and structures” and “Wooden dining tables"). We have updated the Discussion (Section 5) to explicitly highlight how future work could employ hierarchical SAEs.
>
> ---
>
> **> Q3. Concept Dictionary.** We trained an RA-SAE with 32k concepts on DINOv2 features from ImageNet (Appendix F). Unlike previous SAE architectures, the RA-SAE constrains dictionary atoms to lie within the convex hull of the training data, ensuring stability and geometric validity. To map these atoms to human-understandable concepts, we employ an autointerpretability pipeline (described in Appendix F.2). Specifically, we retrieve high-activating exemplars for each concept and use a VLM to generate descriptive labels based on the recurring visual patterns.
>
> ---
>
> We believe the revisions based on this review made the manuscript stronger and more accessible. We would be happy to continue working with the reviewer to confirm that their concerns have been addressed and clear out any outstanding issues. We would also appreciate the reviewer’s reconsideration of the rating if all of the above resolves the raised concerns.

---

> > ### Author Response · Authors · 2025-11-27
> >
> > [ 1 ] Heusel, Martin, et al. "Gans trained by a two time-scale update rule converge to a local nash equilibrium." Advances in neural information processing systems 30 (2017).
> >
> > [ 2 ] Hessel, Jack, et al. "Clipscore: A reference-free evaluation metric for image captioning." Proceedings of the 2021 conference on empirical methods in natural language processing. 2021.
> >
> > [ 3 ] Liu, Qihao, et al. "Discovering failure modes of text-guided diffusion models via adversarial search." arXiv preprint arXiv:2306.00974 (2023).
> >
> > [ 4 ] Fel, Thomas, et al. "Into the Rabbit Hull: From Task-Relevant Concepts in DINO to Minkowski Geometry." arXiv preprint arXiv:2510.08638 (2025).
> >
> > [ 5 ] Ayzenberg, Lev, Raja Giryes, and Hayit Greenspan. "Dinov2 based self supervised learning for few shot medical image segmentation." 2024 IEEE International Symposium on Biomedical Imaging (ISBI). IEEE, 2024.
> >
> > [ 6 ] Cui, Beilei, et al. "Surgical-dino: adapter learning of foundation models for depth estimation in endoscopic surgery." International Journal of Computer Assisted Radiology and Surgery 19.6 (2024): 1013-1020.

---

### Official Review · Reviewer_3MU4 · 2025-11-04

**Soundness:** 3
**Presentation:** 3
**Contribution:** 3
**Rating:** 6
**Confidence:** 2

**Summary:**

The paper introduces a novel framework to systematically discover suppressed or exaggerated conceptual blindspots. In particular, the authors first trained a large sparse autoencoder (RA-SAE) on DINOv2 features to extract an interpretable concept basis. They then define energy differences between the prevalence of each concept in real vs. generated images to pinpoint these blindspots.

**Strengths:**

- The problem is novel and well-motivated, and the paper is well-written.
- While I feel like some of the observations are not new (for instance Sec 4.6), I think it is nice that the framework can be used to quantify them.
- The open-source framework is highly appreciated.

**Weaknesses:**

- Section 4.4 would benefit more from a more systematic study and quantitative results. For instance, when the authors claim that “While some of these discrepancies can be attributed to underspecified or noisy captions, others reveal genuine blindspots”, I think it would be better to quantify what portion of the images with large energy difference has noisy captions.

**Questions:**

- I don’t think the title of Section 4.1 captures its contents? Moreover, some works have shown that T2I models are able to generate unseen concepts [1]

References:
[1] Haviv, A., Sarfaty, S., Hacohen, U., Elkin-Koren, N., Livni, R., & Bermano, A. H. (2024). Not Every Image is Worth a Thousand Words: Quantifying Originality in Stable Diffusion. arXiv preprint arXiv:2408.08184.

---

> ### Author Response · Authors · 2025-11-27
>
> We thank the reviewer for their helpful feedback. We’re glad that the reviewer found the problem to be “novel and well-motivated”, and the paper to be “well-written”. Below, we address the suggestions and questions raised in this review. We have uploaded a revised version of the manuscript, with changes tracked in red.
>
> ---
>
> **> W1. Concretize and Quantify Findings in Sec 4.4: Prompt Noise vs. Genuine Blindspots.** This is a great point. To address this, we conducted a new experiment, where we used a VLM (as an AutoRater / LLM-as-a-judge) to inspect these tail-end, high diverging samples. We found that the majority (56.3% of the 200 most diverging datapoints) are genuine blindspots. We reflected these findings in Section 4.4, and described the experiment in detail in Appendix I. This concretizes the claims, as suggested by the reviewer, and also confirms that the analysis highlights genuine error patterns, not just noisy datapoints.
>
> ---
>
> **> Q1. Title of Sec. 4.1.** We agree with the reviewer that the section title is not optimal, and have changed it to ‘The Heavy Tail of Suppressed Concepts’ to be more consistent and directly descriptive of the findings.
>
> ---
>
> We hope these revisions and clarifications address the reviewer’s concerns, and we would be eager to collaborate further if they have additional suggestions.

---

### Author Response · Authors · 2025-11-27

We thank the reviewers for their time and feedback. We were encouraged that most reviewers found our method to be helpful and insightful, and took time to incorporate their suggestions. We have conducted multiple new experiments, which include:

- a detailed comparison to existing methods (new section, Appendix B);
- an analysis of caption noise in high-divergence datapoints (new section, Appendix I);
- a thorough evaluation of SAE error cases (new section, Appendix J);
- updates to Section 5 (Discussion) with ways of building interventions on top of our framework and generalizing the method to other architectures and contexts.

We revised the manuscript, resulting in **five more pages of appendices**. We tracked the changes in the revised manuscript in red for easier orientation. We believe that these revisions have made the manuscript stronger and more accessible, and we’re looking forward to engaging with the reviewers to clear out any outstanding concerns.

---

### Meta-Review · Area_Chair_GpVu · 2026-01-05

**Summary:**

This paper introduces a rigorous framework for detecting conceptual blindspots in generative image models, which are systematic discrepancies where specific concepts are either under-represented (suppressed) or over-represented (exaggerated) compared to the natural data distribution. The authors employ Sparse Autoencoders (SAEs), specifically a Relaxed Archetypal SAE trained on DINOv2 features with 32,000 concepts, to decompose images into interpretable features and calculate an energy difference metric that quantifies the divergence between real and generated images. Applied to models such as Stable Diffusion, PixArt, and Kandinsky, the analysis reveals distinct failure modes ranging from universal suppressions like whitespaces on documents to model-specific blindspots and memorization artifacts and demonstrates that post-training interventions like DPO can improve conceptual alignment.


The reviewers liked the introduction of a "conceptual blindspot" and thought it was a significant contribution. The also thought Archetypal Sparse Autoencoder (RA-SAE) trained on DINOv2 features, is an interesting, scalable, unsupervised, and mathematically sound. However, several weaknesses were identified, most notably the lack of baseline comparisons to existing methods and the absence of error analysis regarding the SAE's performance (e.g., false positives or redundancies). That said I think the revisions addressed most of these concerns and I think the lowest score might have been increased so I recommend acceptance.

**Reviewer Concerns:**

The authors conducted extensive revisions that directly targeted the primary reasons for lower scores, particularly for Reviewers NSuB and 3MU4.

Reviewer NSuB (Score: 4) had criticized the lack of baselines and error analysis; the authors addressed this by adding a new section comparing their method to others and a detailed error analysis section. Similarly, Reviewer 3MU4 (Score: 6) requested quantification of "genuine" blindspots versus caption noise; the authors responded with a new VLM-based experiment

**Reviewer Scores:**

I think 4 would have likely increases to 6 not sure others would have increased.

---

### Decision · Program_Chairs · 2026-01-26

Accept (Poster)